# Identifying researcher characteristics driving growth in Japanese University-originated deep-tech startups: A machine learning approach

**Yoshifumi Mizuhara**[ID]*, **Tetsusei Kurashiki**

University of Osaka, Osaka, Japan
* mizuhara.yoshifumi.ivs@ecs.osaka-u.ac.jp

## Abstract

Active involvement of researchers with strong outputs is essential for growth of deep-tech university-originated startups. We used the University-Originated Venture Database to link total funding amount with researchers' publication metrics and KAKENHI grant records. Classification models were constructed to distinguish growing and non-growing startups, followed by feature-importance and breakdown-tree analyses to interpret which researcher attributes drive growth. We then applied these defined characteristics to evaluate researcher profiles at top Japanese universities to assess their entrepreneurial engagement. This process revealed a disconnect between high-quality research achievements and actual startup activity. These insights underpin an identification framework that can guide investment and policy decisions in Japan's deep-tech ecosystem.

## 1. Introduction

### Background of the study

In 2022, the Japanese government designated the year as the "Startup First Year," initiating efforts for strengthening the domestic startup ecosystem [1]. A particular focus has been placed on "deep-tech startups," which integrate advanced and innovative technologies into their business models (for example, BioNTech in Germany and Moderna in the US). Such startups often originate from universities and research institutions, a trend that aligns with the Japanese government's approach to bolstering its ecosystem with a university-centered approach. The government aims for deep-tech startups to grow into unicorns or decacorns.

However, to implement this policy effectively, one must recognize that the creation and growth trajectories of deep-tech startups differ fundamentally from other startups.

Firstly, deep-tech startups often arise from research outputs—such as intellectual property, data, and prototypes—developed by university or institutional researchers. Launching such startups therefore requires not only entrepreneurs but also the active involvement of these researchers. As Lerner [2] emphasized, innovation is driven

**Data availability statement:** All data underlying the findings of this study are fully available, except for the proprietary STARTUP DB records, which are available under license from For Startups, Inc., Japan. Researchers may request access to the STARTUP DB proprietary dataset provided by For Startups, Inc. (https://startup-db.com/) by contacting the data provider through the following official web forms: For Startups, Inc. Contact Form at the following link: https://share.hsforms.com/1QbS5NH-8QX-WYQWAisr-Csg556co?__hstc=47617656.6c3897dd7aeeae4add688f0f70b29b17.1762992325682.1763773386915.1764066835595.8&__hssc=47617656.1.1764066835595&__hsfp=3389951778 and STARTUP DB Plan / Estimate Request at the following link: https://lp.startup-db.com/ep/plan?_gl=1*1rbzk-ka*_gcl_au*MTAzMjcxNTMwNi4xNzYyOTky-MzI1). Access is granted at the discretion of For Startups, Inc. and may require a license agreement, similar to the arrangement used for this study. The University-Originated Venture Database (METI) data used in this study are publicly available from the Ministry of Economy, Trade, and Industry and may be accessed here: https://www.meti.go.jp/policy/innovation_corp/univ-startupsdb.html. The KAKENHI grant records are publicly accessible via the KAKEN database (Grants-in-Aid for Scientific Research) here: https://kaken.nii.ac.jp/. The Scopus metadata is available from Elsevier Scopus (an institutional subscription is required): https://www.elsevier.com/solutions/scopus. Each Scopus Author ID used in this study can be accessed directly via the following URL format: https://www.scopus.com/authid/detail.uri?authorId=XXXXXXX. The Supporting information file S1_Dataset_Linkage.xlsx provides the complete list of Scopus Author IDs in this URL-accessible format. The relevant eradIDs used in the study are also provided in S1_Dataset_Linkage.xlsx. The Supporting Information file "S1_Dataset_Linkage.xlsx" also contains the full linkage table used in this study, including company identifiers, STARTUP DB IDs, Scopus Author IDs (URL-accessible), and KAKEN eradIDs, ensuring reproducibility.

**Funding:** The author(s) received no specific funding for this work.

not merely by institutional reforms but by effective mechanisms that connect academic research, venture capital, and entrepreneurship, highlighting the pivotal role of researchers in transforming scientific knowledge into economic value. Likewise, Visintin and Pittino [3] demonstrated that university-based spin-offs with founding teams combining academic and managerial expertise outperform those composed solely of academics, underscoring that researchers' active participation is critical for early-stage growth.

Secondly, for deep-tech startups to achieve sustainable growth, they require longer development cycles and substantial funding from the outset. Unlike lean startup methods [4], the path for deep-tech startups is quite different [5]. For instance, Boston Consulting Group [6] reports that, while deep-tech startups represent a small portion of the overall startup ecosystem, investments in them surged from $15 billion in 2016 to over $60 billion in 2020 [7]. Additionally, these startups now constitute around 20% of venture capital investments, up from about 15% a decade ago [8]. According to EUROPEAN STARTUPS [9], deep-tech startups in Europe, particularly in biotechnology, generally experience longer intervals between funding rounds than other startup types.

In summary, deep-tech startups require researchers, substantial funding and extended development periods from their inception to achieve sustainable growth. Consequently, investing domestic resources in their growth involves higher risk compared to other types of startups, calling for careful consideration in investment decisions. Hence, assessing researchers' outputs and their profiles at the early stage is crucial for investment decisions.

Deep-tech startups are typically born out of academic research, requiring not only advanced scientific knowledge but also mechanisms that enable effective knowledge transfer from universities to industry. Lockett et al. [10] conceptualized such university-based spin-offs within a knowledge-based view, defining them as independent firms created to commercialize publicly funded research and emphasizing that successful commercialization depends on bridging "knowledge gaps" between scientific and entrepreneurial capabilities. Meyer [11] further highlighted that researchers play diverse roles in the commercialization of academic knowledge, distinguishing between "academic entrepreneurs," who establish firms to pursue the growth of research-based ventures, and "entrepreneurial academics," who remain primarily research-oriented while engaging in commercialization. He argued that public support mechanisms can shape these behaviors, emphasizing that managerial expertise and external networks are crucial for successful university spin-offs. Marx and Hsu [12] focused on the phenomenon of similar scientific papers being published in close succession, creating a sample set of "Twin" papers. Their analysis revealed that research outcomes proposed in Twin studies tend to ensure scientific progress and competitive advantage, thereby more easily leading to commercialization by deep-tech startups. Additionally, Goji et al. [13,14] analyzed the relationships between researchers in specific scientific fields and deep-tech startups, finding that central researchers in these fields are more likely to participate in deep-tech startups. Thus, the core research outcomes and the

researchers behind them are pivotal for deep-tech startups to successfully secure development funds and achieve continuous growth.

Recognizing that deep-tech startups, like other startups, fundamentally depend on skilled management teams and a supportive ecosystem to thrive, Japan should additionally focus on identifying researchers with research outcomes that can drive such growth and connecting them with entrepreneurial opportunities. However, Japan's startup ecosystem remains relatively small on a global scale, with venture capital-backed cases in 2021 totaling 1,915—about one-eighth of the United States' total and approximately two-fifths of China's [15]. Therefore, to foster more high-growth deep-tech startups within Japan, it is essential and efficient to identify and support researchers and research outcomes with strong potential for growth. As Mazzucato [16] argued, because the state itself can act as a primary risk-taking agent in innovation, this approach is particularly relevant for Japan.

## Purpose

Given Japan's limited number of startups, establishing deep-tech startups involving researchers with outstanding research achievements is an efficient means to generate more unicorns or decacorns. This requires first identifying "researchers with outstanding achievements that contribute to growth" (hereafter referred to as "involved researchers"). It then requires initiating discussions around their precise definition and the development of an identification model. We therefore begin by analyzing actual growth outcomes of Japanese deep-tech startups and the characteristics of their involved researchers. Since prior works by Hsu and Goji rely on global datasets, directly applying their findings to Japan—where research funding environments and startup economics differ—could lead to imprecise assessments. Those studies primarily use publication metrics to evaluate researchers and define growth simply as "startup founded," which may not align with Japan's context.

Therefore, in this study we utilize Japan-specific data to analyze the relationship between deep-tech startup growth and the characteristics of involved researchers, and to develop an identification model. We focus on university-originated startups not because we assume systematic differences from other Japanese deep-tech startups, but because reliable, government-maintained datasets exist only for the university-originated subset, making it the sole nationally standardized source for analysis in Japan.

This study will actively incorporate machine learning techniques into the analysis. The datasets of startups and researchers contain multivariate data with various parameters. Machine learning is a powerful tool for analyzing such multivariate data, and its application has recently become more prominent in startup research. For instance, Zbikowski et al. [17] developed a machine learning model to predict business success using the Crunchbase database, which focuses on U.S.-based startups. Additionally, Cao et al. [18] discussed the use of deep learning (DL) to predict startup success, while Qian et al. [19] developed a model linking company information, region, and venture capital funding to successful outcomes, such as acquisitions and IPOs, utilizing machine learning techniques. These examples demonstrate the increasing use of machine learning in startup analysis, and this study aims to develop an efficient identification model using similar techniques. In particular, we apply an AutoML tool to efficiently construct interpretable tree-based classification models.

Building on this foundation, the present study quantitatively analyzes the relationship between the growth of university-originated startups in Japan and the involved researchers, with the goal of constructing an identification model for research outputs and researcher characteristics that contribute to the growth of deep-tech startups. Specifically, we first collect data on the growth performance of Japanese university-originated startups, along with publication records and KAKENHI (Grant-in-Aid for Scientific Research) information of the associated researchers. Next, machine learning techniques are applied to the collected data to develop and evaluate a classification model that estimates startup growth based on the research outputs of the involved researchers. Finally, based on the model evaluation results, we identify researcher characteristics that significantly contribute to the growth of deep-tech startups. These characteristics are then

used to compare and analyze the performance of researchers at universities that rank highly in the number of university-originated startups they produce.

## 2. Data collection

To analyze the relationship between the growth of Japanese university-originated startups and the characteristics of the involved researchers, a reliable data list is essential. The "University-Originated Venture Database", conducted by the Ministry of Economy, Trade and Industry [20]. This database contains information on university-affiliated startups, such as business descriptions, locations, involved researchers, affiliated universities, and technological fields. Our analysis used the version released on June 2, 2023.

We supplemented financing and other startup attributes via STARTUP DB (For Startups, Inc.), a leading domestic platform covering over 20,000 Japanese startups with comprehensive financial and business data. Limited data access is free; full access requires a paid subscription.

To gather additional data on the research publications and funding details of involved researchers, the "Scopus" and "KAKEN" databases were used. "Scopus," managed by Elsevier, is a widely-used international abstract and citation database that also offers information on co-authors, making it used in universities and research institutions. For additional details on journals, metrics like the Journal Impact Factor (JIF), published by Clarivate Analytics' Journal Citation Reports, are also used. "KAKEN," managed by the Japan Science and Technology Agency (JST), is an open-access system providing data on research projects funded by the Grants-in-Aid for Scientific Research Program (KAKENHI). It includes information on budgets, project duration, principal researchers, and collaborators.

## 3. Data cleaning/ setting variables

### Data cleaning

Fig 1 summarizes the data integration and cleaning process used to construct the final analytical sample.

The "University-Originated Venture Database" contained 967 listings, which were matched with "STARTUP DB" using corporate numbers as the primary identifier. As a result, 357 startups were successfully linked between the two databases. A total of 610 entities were excluded at this stage. This reduction reflects structural differences in database definitions rather than data loss. Specifically, the University-Originated Venture Database maintained by the Ministry of Economy, Trade and Industry (METI) allows the inclusion of a broad range of organizational forms, such as general incorporated associations, and does not require listed entities to have conducted external fundraising. In contrast, STARTUP DB focuses on venture companies with identifiable corporate registration and financial activity, resulting in a narrower and more investment-oriented subset.

To examine funding trends over time, these 357 companies were divided into two groups based on their founding years, and statistical significance was tested. With 2015 identified as a key threshold year ($p < 0.05$), only startups founded from 2015 onward (194 companies) were selected for analysis, as illustrated in Fig 1. This year corresponds to a major institutional and financial shift initiated by the enforcement of the Industrial Competitiveness Enhancement Act (April 1, 2014) and the launch of the Public-Private Innovation Program by the Ministry of Education, Culture, Sports, Science and Technology (MEXT). Under this program, four major national universities—the University of Tokyo, Kyoto University, Osaka University, and Tohoku University—each established university-affiliated venture capital funds with a total investment capacity of approximately 100 billion JPY, beginning in FY 2014. This policy context does not imply that our analytical sample was restricted to these four universities; the startups in the final dataset originated from a broader range of universities. Therefore, the post-2015 subset represents startups founded under this new policy and funding environment, ensuring analytical consistency within a unified institutional context [21].

Among the 194 companies analyzed, a total of 252 involved researchers were identified, indicating that some startups are associated with more than one researcher. For model construction and evaluation, we did not aggregate

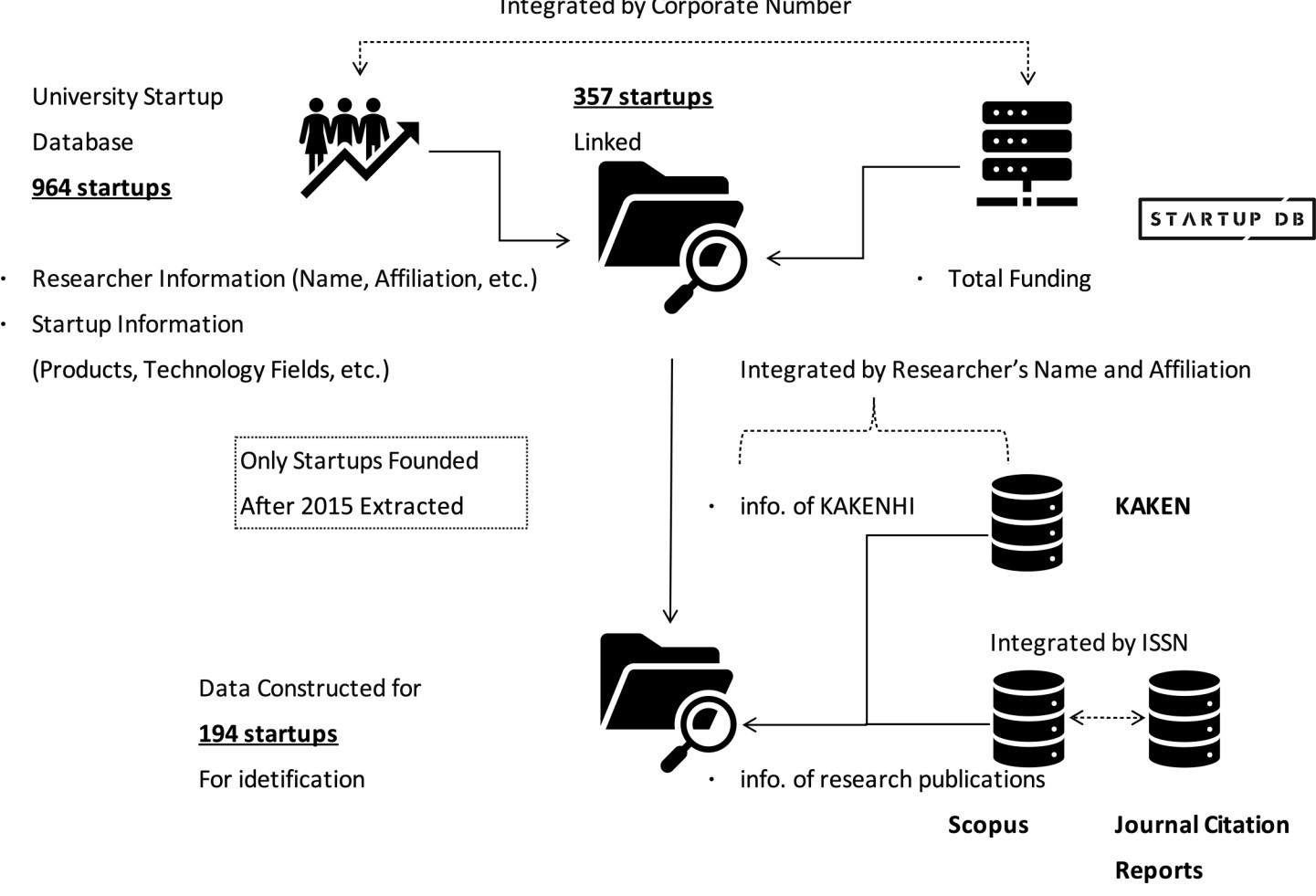

**Fig 1. Relationships Between Each Dataset.**

researcher-level metrics within a startup (e.g., by simple or weighted averages); instead, we created a one-to-one startup–researcher dataset by selecting one researcher per startup via repeated random sampling, as described below.

## Target variables

We classified university-originated startups into Positive (growing) and Negative (not growing) groups using total funding amounts from STARTUP DB. To achieve this, we linked the University-Originated Venture Database with STARTUP DB.

Additionally, to refine our identification, we filtered startups by the main technology field of their core product or service as listed in the Venture Database. The thresholds for funding amounts and technology fields are summarized in Table 1.

We imposed a time cutoff to ensure that only information available by the startup's founding year was used. For KAKENHI, we collected projects up to 10 years before foundation. For publications, we retrieved two datasets: (i) all papers before the founding year and (ii) papers from the five years immediately preceding foundation to reduce bias from cumulative achievements.

For certain predictor variables in Table 2 that are not directly output from databases, calculations were made post-data retrieval. Specifically, variables related to research delegation, contracted research, collaborative research in the KAKEN

**Table 1. Example of table.**

| | Case1 | Case2 | Case3 | Case4 |
|---|---|---|---|---|
| Positive | ≧1JPY | ≧500MJPY | ≧1JPY | ≧1JPY |
| Negative | 0JPY | 0JPY | 0JPY | 0JPY |
| Technology Field | All | All | Biomedical | Non-Biomedical |
| Number of Verification Data/ Remaining Data | 95 x 2/ Positive: 4 | 51 x 2/ Negative: 44 | 33 x 2/ None | 62 x 2/ Positive: 4 |

*For Positive/Negative, threshold conditions were set based on the target variable, which is the amount of funding raised. Startups that met the threshold were classified as Growing＞Positive, while those that did not meet the threshold were classified as Not Growing＞Negative. Startups that did not fall into either category were not used in the analysis. Although the nominal threshold of "1 JPY" may appear low, in practice, the companies that received any external funding exhibited substantially high financing levels. Among the 99 companies with total funding ≥1 JPY, the mean amount was 1,048 million JPY and the median was 559 million JPY. Only 16 companies raised less than 100 million JPY, and 3 companies less than 10 million JPY. Therefore, this criterion effectively separates startups that have successfully secured significant external funding from those that have not.

*The remaining data refers to the group and count of data left after aligning the number of records with the smaller group in the two-group analysis.

* Startups were filtered using the METI University-Originated Venture Database category "main technology field related to the core product/service." In the Bio subset, we retained only firms classified as "Bio & Healthcare" and excluded other categories

data, and co-author information in the publication data were compiled into separate "Collaborative Research Data" and "Co-Author Data" sets. In the "Collaborative Research Data," peers who collaborated on the same project (hereafter, collaborative researchers) were extracted from KAKEN, and co-authors were identified from Scopus, ensuring no duplication with the primary researcher. The "Co-Author Data" included the frequency of co-authorship per pair, and the "Collaborative Research Data" recorded the project counts and total research budgets, categorized by project role (principal researcher, collaborative researcher, or otherwise). Total counts of co-authors and collaborative researchers were then calculated for each primary researcher.

### Predictor variables

Predictor variables consisted of researchers' publication metrics (e.g., citation counts, JIF scores) from Scopus and their KAKENHI funding information (e.g., total budget, number of projects). Table 2 lists all selected variables.

## 4. Machine learning model construction, evaluation, and interpretation

### Construction and evaluation method of the machine learning model

We used the Tree-Based Pipeline Optimization Tool (TPOT) to build a classification model relating the target variables (Table 1) to the predictor variables (Table 2). TPOT is an open-source AutoML tool that automates pipeline development [22]. We selected TPOT because it is open source—facilitating easy setup of our research environment—and because it integrates seamlessly with Python's scikit-learn library. While other AutoML tools may outperform TPOT in some respects, TPOT offers a good balance between cost-effectiveness and system requirements.

TPOT's flexibility in choosing interpretable tree-based algorithms is a key advantage. Although deep learning models can achieve high accuracy, they often obscure the relationships between target and predictor variables. TPOT can incorporate deep learning methods, but this study prioritized interpretability and thus limited algorithm selection to the tree-based methods listed in Table 3.

To construct a consistent one-to-one dataset for modeling, one researcher was randomly selected per startup. This random sampling was repeated five times, and the average model performance across these iterations was used to evaluate the model's stability. In each iteration, the sampled dataset contains at most one researcher per startup; therefore, no within-startup averaging of researcher metrics was performed. This approach ensured that model evaluation did not depend on any particular random draw of researchers from startups with multiple participants.

**Table 2. Predictor variables.**

| # | Variable Name | Source | Type | Overview | Remark Remarks |
|---|---|---|---|---|---|
| 1 | Number of Research Projects | K | N | Number of projects obtained as PR | 3 |
| 2 | Funding Amount | K | N | Total budget for projects obtained where the researcher is the PR | 3 |
| 3 | Maximum Funding Amount | K | N | Maximum budget per project obtained where the researcher is the PR | 3 |
| 4 | Average Funding Amount | K | N | Average budget per project obtained where the researcher is the PR | 3 |
| 5 | Total Number of Commissioned Research Projects | K | N | Number of times the researcher, as PR, commissioned to co-researchers | 2 |
| 6 | Total Number of Participations in Co-Rs' Projects | K | N | Number of times the researcher participated in projects where co-researchers are the PRs | 2 |
| 7 | Total Number of Participations in other Projects | K | N | Number of times the researcher participated in projects where neither the researcher nor co-researchers are PRs | 2 |
| 8 | Total Funding of Participations in Co-Rs' Projects | K | N | Total budget for #6 | 2 |
| 9 | Total Funding of Participations in other Projects | K | N | Total budget for #7 | 2 |
| 10 | Total Number of commissioned Co-R Count | K | N | Number of commissioned co-researchers in #5 | 2 |
| 11 | Number of Commissioned Researchers | K | N | Number of co-researchers in #6 | 2 |
| 12 | Number of Collaborative Researchers | K | N | Number of co-researchers in #7 | 2 |
| 13 | Total Number of Collaborative Participants | K | N | Number of co-researchers in #5~7 | 2 |
| 14 | Total Research Expense | K | N | Total budget in #2, 8 and 9 | 2 |
| 15 | h-index | K | N | h-index | |
| 16 | Avg Documents Per Year | K | N | Average Annual Number of Publications | 3 |
| 17 | Document Count (All years) | S | N | Total Number of Publications (All years) | |
| 18 | Cited By Count (All years) | S | N | Total Number of Citations (All years) | |
| 19 | Max Cited By (All years) | S | N | Maximum Number of Citations for a Single Paper (All years) | 3 |
| 20 | Avg Cited By (All years) | S | N | Average Number of Citations per Paper (All years) | 3 |
| 21 | Max JIF (All years) | J | N | Maximum Journal Impact Factor of Published Papers (All years) | 3 |
| 22 | Avg JIF (All years) | J | N | Average Journal Impact Factor of Published Papers (All years) | 3 |
| 23 | Max JCI (All years) | J | N | Maximum Journal Citation Indicator of Published Papers (All years) | 3 |
| 24 | Avg JCI (All years) | J | N | Average Journal Citation Indicator of Published Papers (All years) | 3 |
| 25 | Total Extended Count of Co-Authors | S | N | Total Number of Co-Authors (All years) | 3 |
| 26 | Total Number of Co-Authors | S | N | Total Number of Unique Co-Authors (All years) | 3 |

*(Continued)*

**Table 2.** (Continued)

| # | Variable Name | Source | Type | Overview | Remark Remarks |
|---|---|---|---|---|---|
| 27 | Document Count(5 years) | S | N | Total Number of Publications(5 years) | |
| 28 | Cited By Count (5 years) | S | N | Total Number of Citations(5 years) | |
| 29 | Max Cited By (5 years) | S | N | Maximum Number of Citations for a Single Paper (5 years) | 3 |
| 30 | Avg Cited By (5 years) | J | N | Average Number of Citations per Paper (5 years) | 3 |
| 31 | Max JIF (5 years) | J | N | Maximum Journal Impact Factor of Published Papers (5 years) | 3 |
| 32 | Avg JIF (5 years) | J | N | Average Journal Impact Factor of Published Papers (5 years) | 3 |
| 33 | Max JCI (5 years) | J | N | Maximum Journal Citation Indicator of Published Papers (5 years) | 3 |
| 34 | Avg JCI (5 years) | J | N | Average Journal Citation Indicator of Published Papers (5 years) | 3 |
| 35 | Institution | K | C | | |

\* Source; Data Source, K – KAKEN, S – SCOPUS, J- Journal Citation Reports.

\* Type; Variable type, N – Numeric (Continuous Variable), C – Categorical Variable.

\* Overview; PR – Principal Researcher, (All years) – Entire Research Period Before Establishment, (5 years) – 5 Years Before Establishment).

\* Remark; 1 – Calculated from Data Source, 2 – Create unique one-to-one pairs of researchers and co-researchers in Research Projects without duplication, defining co-researchers as 'Co-Researchers.' Then, calculate based on the definition from the Data Source, 3 – Create unique one-to-one pairs of researchers and their co-authors in Research Projects without duplication, defining co-authors as 'Co-Authors.' Then, calculate based on the definition from the Data Source, 4 – Use in analysis by assigning '1' to the University of Tokyo, Kyoto University, Osaka University, and Tohoku University, and '0' to all others.

We evaluated model performance using 5-fold cross-validation: in each fold, the model was trained on 80% of the data and evaluated on the held-out 20%, and we report the mean of the evaluation metrics across folds. We report F1 score and ROC-AUC as the primary complementary metrics, while Accuracy is provided to support interpretability. To mitigate class imbalance, we downsampled the larger group to match the smaller group's size via random sampling. This adjustment was applied in Cases 1, 2, and 4, as the number of Negative (non-growing) startups exceeded that of Positive (growing) ones, as indicated in the "Number of Verification Data/ Remaining Data" row in Table 1. Feature importance scores were computed to assess each predictor variable's influence.

**Model evaluation results**

Following the criteria in Cases 1–4 outlined in Table 1, four classification models were constructed and evaluated for accuracy using TPOT. The model performance metrics for Cases 1–4 are presented in Table 4, feature importance scores are illustrated in Fig 2.

As shown in Table 4, the model's performance improved as the distinction between the compared groups became clearer. Accuracy and F1 score increased when the funding threshold was raised from Case 1 to Case 2, indicating that a larger gap in funding scale enhanced the model's ability to discriminate between growing and non-growing startups. Furthermore, when the dataset was restricted to specific technology-field subsets (Cases 3 and 4; biomedical vs. Non-Biomedical), both accuracy and AUC were higher than in the non-restricted cases, suggesting that analyzing more homogeneous technology domains contributed to improved classification performance. However, this binary split should be interpreted as a pragmatic, exploratory adjustment under limited sample size and should not be regarded as a statistically sufficient control for broader sectoral heterogeneity; future work with larger samples should incorporate more granular technology-sector controls.

**Table 3. Example of table.**

| # Feature Selectors |
| --- |
| sklearn.feature_selection.SelectFwe |
| sklearn.feature_selection.SelectPercentile |
| sklearn.feature_selection.RFE |
| sklearn.feature_selection.SelectFromModel |
| # Feature generators |
| sklearn.preprocessing.PolynomialFeatures |
| sklearn.preprocessing.Binarizer |
| sklearn.preprocessing.MaxAbsScaler |
| sklearn.preprocessing.MinMaxScaler |
| sklearn.preprocessing.Normalizer |
| sklearn.preprocessing.RobustScaler |
| sklearn.preprocessing.StandardScaler |
| tpot.builtins.ZeroCount |
| tpot.builtins.OneHotEncoder |
| # Classifiers |
| sklearn.tree.DecisionTreeClassifier |
| sklearn.ensemble.ExtraTreesClassifier |
| sklearn.ensemble.RandomForestClassifier |
| sklearn.ensemble.GradientBoostingClassifier |
| xgboost.XGBClassifier |

**Table 4. Model performance metrics for Cases 1 to 4.**

|  | *Case1* | *Case2* | *Case3* | *Case4* |
| --- | --- | --- | --- | --- |
| Accuracy | 0.67 | 0.78 | 0.81 | 0.78 |
| F1 | 0.74 | 0.80 | 0.82 | 0.79 |
| AUC | 0.66 | 0.73 | 0.75 | 0.82 |

Regarding feature importance, Fig 2 shows that a few predictor variables consistently recorded high scores across all cases. This indicates that a limited number of predictor variables predominantly influence the model's classification decisions.

## Creation of breakdown trees from the machine learning model

Based on the feature importance results shown in Fig 2, both commonalities and differences were observed in the predictor variables emphasized by the model across each case. It should be noted that the feature importance scores presented here represent statistical associations identified by the model and do not imply direct causal relationships between researcher characteristics and startup growth. Therefore, we discuss these variables as factors that most influence the classifier's decisions (feature importance), not as causal determinants of startup growth.

In all cases, variables derived from KAKENHI data, particularly Funding Amount, Maximum Funding Amount, Average Funding Amount, and Number of Research Projects, consistently exhibited high importance scores. These findings suggest that the scale and continuity of research funding are fundamental factors for distinguishing growing startups.

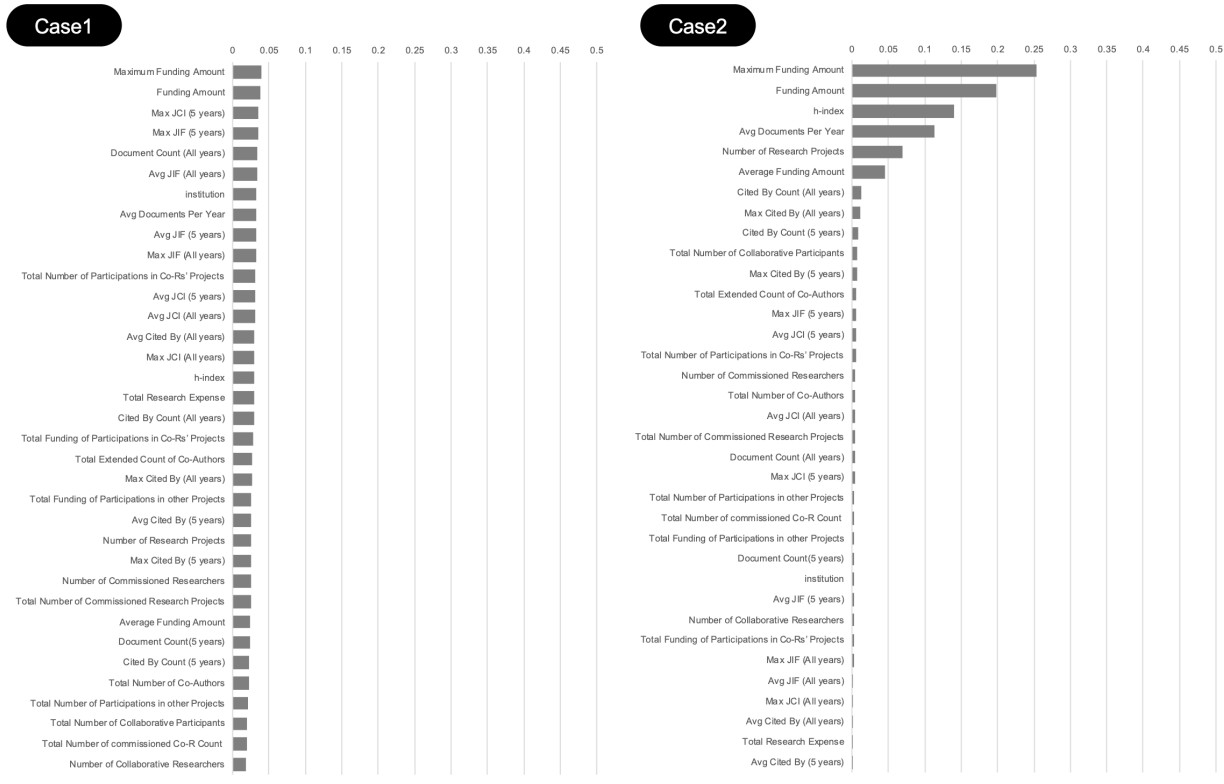

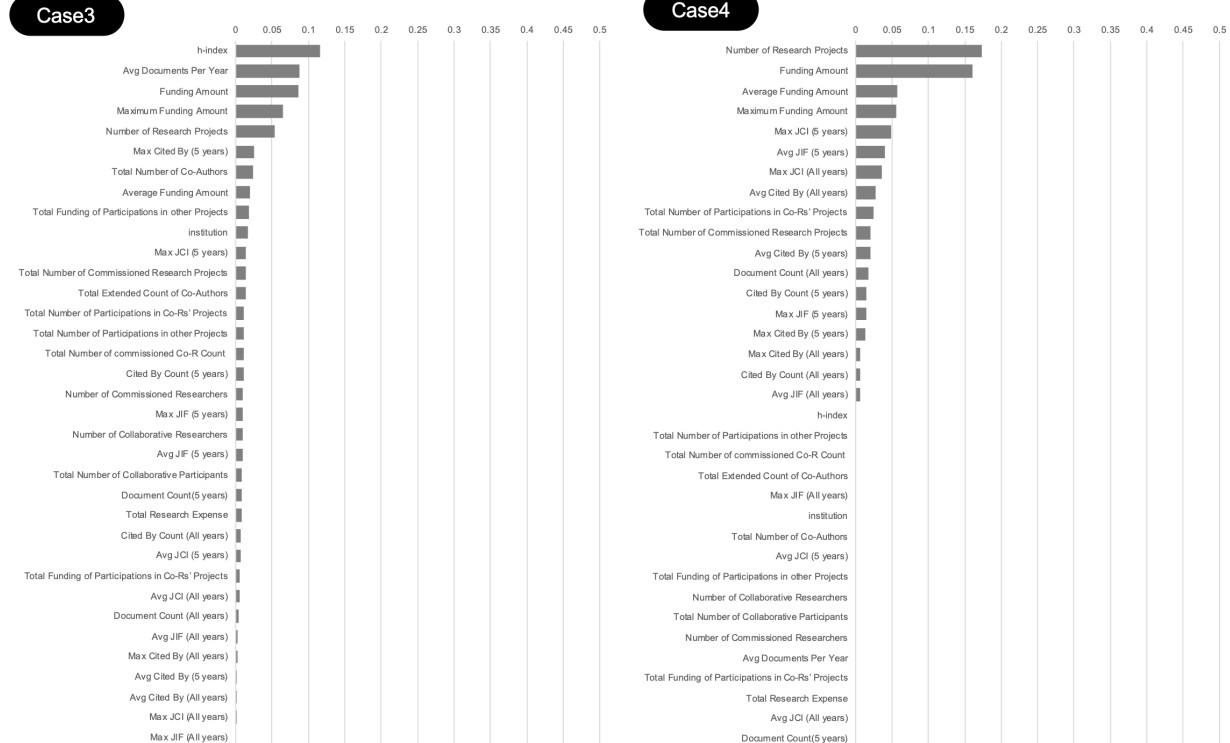

**Fig 2. Feature Importance of Models Built in Cases 1 to 4.**

In addition to these funding-related variables, publication-related metrics—especially quality-oriented indicators such as Max JIF (5 years) and Avg JIF (5 years), alongside citation counts and h-index—also contributed to the models, with the prominence of these quality-oriented indicators becoming more apparent in the field-stratified analyses (Cases 3 and 4).

Fig 3 presents examples of violin plots for predictor variables, showing that Positive startups tended to exhibit higher values than Negative ones, particularly in variables related to research funding and publication quality. This visualization indicates that researchers associated with Positive startups generally possess both stronger funding records and higher-quality publication profiles.

These variables also showed strong mutual correlations, as indicated by the Spearman correlation coefficients, and their detailed relationships are illustrated in Fig 4 (Correlation Matrix). Because Funding Amount, Maximum Funding Amount, and Average Funding Amount are closely related measures derived from the same underlying KAKENHI funding records, the strong correlations among these variables are expected and primarily indicate redundancy/overlap among related funding indicators (i.e., potential multicollinearity), rather than any "quality" implication. Publication-related metrics also showed relatively high internal correlations; in particular, citation- and journal-based indicators (e.g., Max/Avg JIF (5 years), JCI-related variables, and h-index) are conceptually related measures of research excellence, so stronger correlations among them are expected and should be interpreted as overlap and potential multicollinearity rather than as evidence of "quality." These indicators were less strongly correlated with quantity-oriented measures such as the number of papers or number of co-authors.

Focusing on these results, breakdown trees were created for each case, as shown in Fig 5, to analyze the internal structure of the model. To enhance interpretability, all cases were examined with particular attention to Funding Amount and Max JIF (5 years), which consistently appeared among the top predictor variables. For the breakdown-tree visualization only, when multiple researchers were associated with a single startup, we selected a single representative researcher per startup to facilitate interpretation of the tree paths. Specifically, we chose the researcher with larger values for the Positive class and the researcher with smaller values for the Negative class to create a clear one-to-one mapping for illustrative purposes. This heuristic was used solely to improve interpretability of the breakdown trees and was not used for model training or performance evaluation, which relied on repeated random sampling as described above.

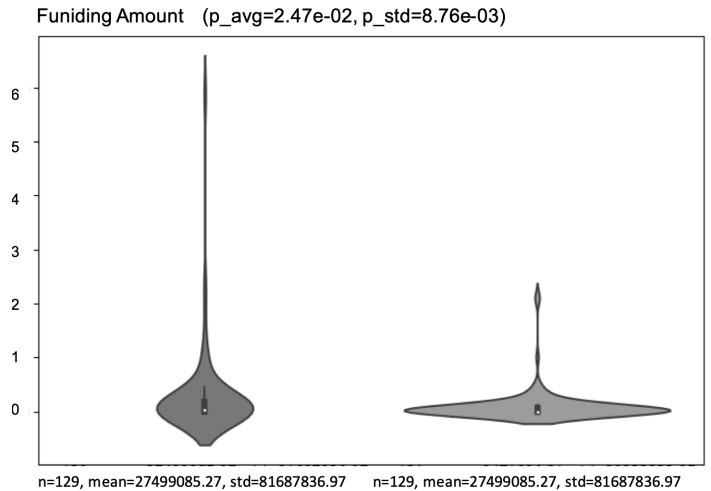
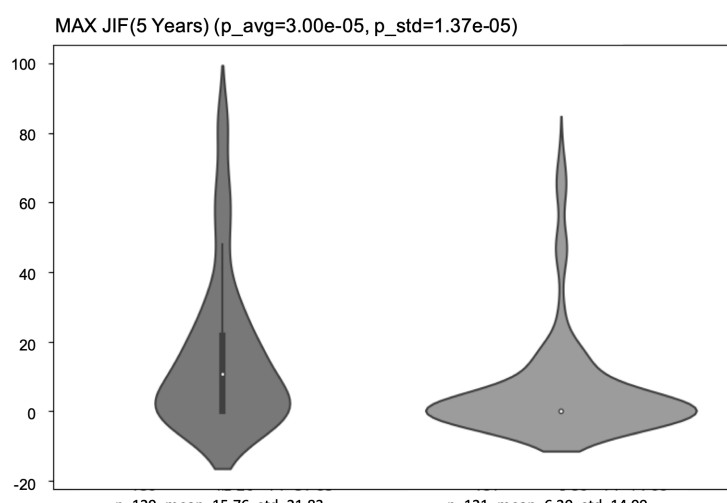

**Fig 3. Examples of violin plots for predictor variables.**

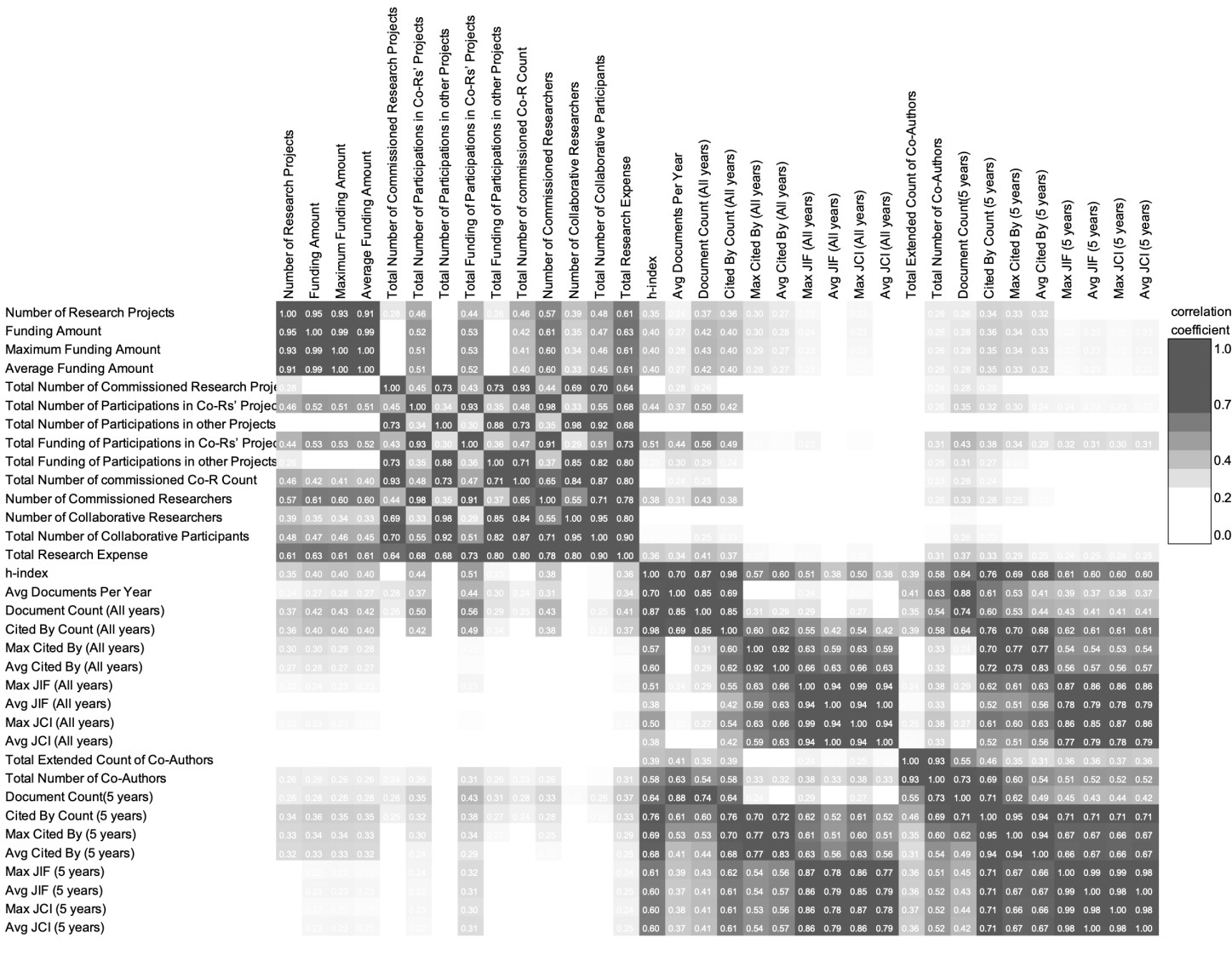

**Fig 4. Correlation Matrix.**

In Case 1, the initial split was on Funding Amount: of 49 cases with ≥10 million JPY, 45 (≈92%) were Positive. The next split used Max JIF (5 years): among the remaining, 41 of 45 with MAX JIF ≥ 10 were Positive. The rest were classified by more complex combinations.

In Case 2, Funding Amount was again the primary discriminator: researchers associated with startups raising more than 15 million JPY were classified as Positive in ≈ 89% of cases. The next most influential variable was Max JIF (5 years) ≥ 13, identifying a further subset of Positive entries.

In Case 3, corresponding to the biomedical domain, the initial split was based on Max JIF (5 years) ≥ 20, which correctly classified the majority of Positive entries (≈ 82%). Among the remaining data, Funding Amount > 20 million JPY further identified additional Positives, indicating that both publication quality and research funding jointly contributed to classification in this field.

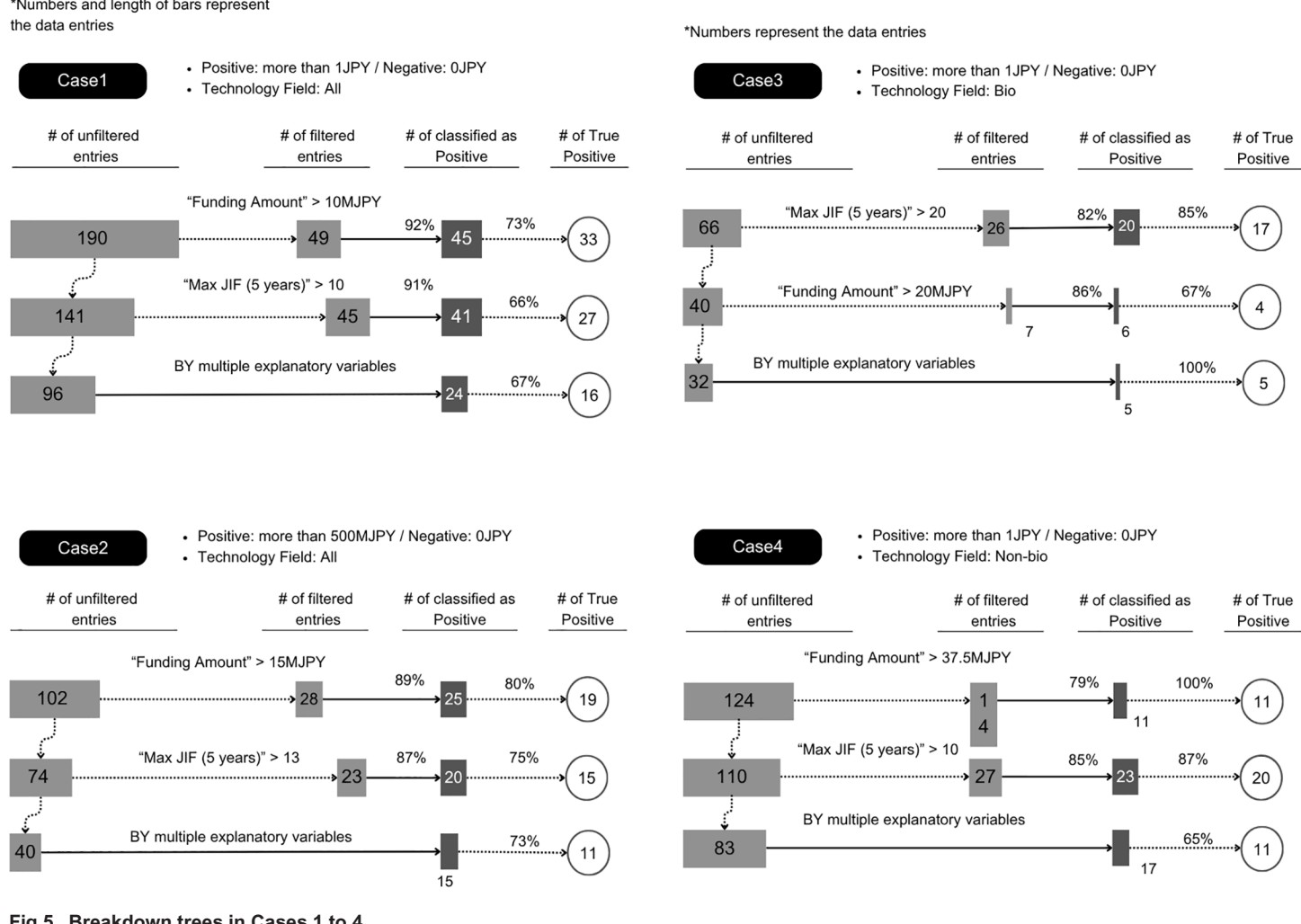

**Fig 5. Breakdown trees in Cases 1 to 4.**

In Case 4, representing the Non-Biomedical domain, the first split used Funding Amount > 37.5 million JPY, followed by Max JIF (5 years) > 20. Researchers satisfying both conditions were classified as Positive in about 80% of cases, highlighting the combined effect of substantial research funding and high-impact publications.

In summary, across all cases, the model performed stepwise classification primarily based on quantitative variables related to Funding Amount and Max JIF (5 years), which were consistently defined as the first or second split in every case. Within the model framework, these two variables were central to predicting startup growth (i.e., to the model's classification decisions). Such threshold-based separations also enhanced the interpretability and transparency of the decision process.

## Logistic regression analysis for interpretation

To support interpretation of the machine-learning results, we additionally conducted logistic regression analyses using the binary growth labels defined for each case (Table 1). This complementary analysis was performed to assess the statistical significance of key predictor variables identified by the ML models.

## Model specification and evaluation procedure

For each analysis case (defined below), the target variable was the binary startup growth label as defined in Table 1 (1 = positive/growing; 0 = negative/non-growing). The set of predictor variables followed the definitions described in Section 3.3. To keep this complementary analysis consistent with the ML interpretation, we screened candidate predictor variables based on the feature-importance ranking (Fig 2) and focused on a parsimonious subset of **up to three** predictor variables. To mitigate multicollinearity, we further removed highly correlated variables by referring to the correlation structure shown in Fig 4; therefore, the **final number of predictor variables could differ across cases**, especially in field-stratified subsets.

Because the dataset construction involved random selection steps (e.g., when multiple researcher profiles could be linked to a startup), we adopted a repeated sampling strategy to assess robustness. Specifically, we repeated the sampling-and-estimation process R = 100 times. Within each repetition, we performed 5-fold cross-validation to evaluate predictive performance and to reduce sensitivity to a single train/test split. The primary performance metric was the area under the receiver operating characteristic curve (AUC). We report the distribution of AUC across repetitions using the median and interquartile range (IQR), as well as the 2.5th and 97.5th percentiles.

Within each cross-validation fold, continuous predictor variables were standardized (z-score) using statistics estimated from the training data and then applied to the corresponding test data to avoid information leakage. In addition, to assess statistical significance for the selected predictor variables, we fitted a logistic regression model to the dataset used within each repetition (after the sampling step) and report odds ratios (ORs) with 95% confidence intervals and p-values. We summarize the OR estimates across repetitions using the median, IQR, and the 2.5th–97.5th percentile range.

## Results of complementary logistic regression analyses

Following the case definitions in Table 1, we conducted complementary logistic regression analyses using the same binary target variable as in the ML classification. Model performance was evaluated under repeated sampling (R = 100) with 5-fold cross-validation, and the ROC–AUC distribution was summarized using the median, interquartile range (IQR), and the 2.5th–97.5th percentile range.

Overall, logistic regression achieved moderate discrimination in the all-field analyses (Cases 1–2), while performance decreased in the Biomedical-stratified analysis (Case 3) and remained moderate in the Non-Biomedical-stratified analysis (Case 4) (Table 5). This indicates that part of the separability captured by the ML models can be reproduced by an interpretable linear log-odds model, but that predictive capacity differs across cases and modeling approaches. In particular, the ML models showed higher discrimination in the field-stratified settings (Table 4), suggesting that non-linear relationships and/or interactions captured by the ML approach contribute to improved classification performance beyond a linear specification.

To facilitate interpretability and to align this analysis with the ML interpretation, we focused on a reduced set of predictor variables that were emphasized by the ML feature-importance results (Fig 2), while considering the correlation structure among researcher indicators (Fig 4) when finalizing the predictor variable set. We report odds ratios (ORs) with

**Table 5. Logistic regression performance (AUC) across repeated sampling (R = 100).**

| Case (See Table 1) | AUC (median) | AUC (p25–p75) | AUC (p2.5–p97.5) |
|---|---|---|---|
| Case 1 | 0.667 | 0.655–0.674 | 0.637–0.691 |
| Case 2 | 0.680 | 0.670–0.691 | 0.632–0.724 |
| Case 3 | 0.566 | 0.528–0.598 | 0.453–0.644 |
| Case 4 | 0.657 | 0.644–0.668 | 0.614–0.685 |

95% confidence intervals and summarize their distributions across repetitions (Table 6). Across cases, predictor variables reflecting research excellence (e.g., publication-quality metrics such as JIF-related indicators, citation-based indicators, and h-index) tended to show positive associations with the probability of being classified as Positive, consistent with the ML models' emphasis on a limited number of high-importance variables.

However, statistical significance was not uniformly stable across repetitions for all predictor variables. This is expected given the limited effective sample sizes (especially in stratified cases) and the multicollinearity among indicators that capture similar conceptual dimensions of researcher excellence (Fig 4). Accordingly, we interpret the logistic regression results as complementary evidence of association—providing effect-size summaries and a robustness check on directional consistency—rather than as estimates of causal effects.

## 5. Consideration of researcher characteristics associated with the growth of university-originated startups

### Definition of researcher characteristics associated with startup growth

The breakdown tree and feature-importance analyses (Cases 1–4) indicated that two major groups of variables characterize researchers associated with startup growth: (1) research funding–related factors, particularly Funding Amount, Maximum Funding Amount, and Average Funding Amount derived from KAKENHI data; and (2) publication-quality indicators, such as Max/Avg JIF (5 years), JCI-related variables, and h-index.

These variables consistently ranked highest in the model and appeared as the primary splitting factors in the breakdown trees. Based on these findings, we discuss the researcher characteristics that contribute to startup growth, focusing on Funding Amount and publication-quality metrics.

Among the KAKENHI-related variables, Funding Amount had the highest feature importance. This variable refers to the cumulative amount of KAKENHI received as a principal investigator, and it was found that cumulative funding of over 10 million yen is a strong positive characteristic. In contrast, other funding-related variables, such as the number of co-investigators or the amount of commissioned research funding, had minimal impact on the classification model. These results suggest that experience in leading research projects—as evidenced by principal investigator achievements—is particularly emphasized.

As for publication-related variables, those representing the quality of research outputs—such as Max/Avg JIF (5 years), JCI-related indicators, and h-index—were found to be important across all case analyses. In particular, these quality-oriented variables contributed notably to classification performance in Cases 3 and 4, where the data were stratified by

**Table 6. Logistic regression results (odds ratios) across repeated sampling (R = 100).**

| Case (See Table 1) | Predictor variables | OR (median) | OR (p2.5–p97.5) | sig_rate_0.05 | mean_p |
|---|---|---|---|---|---|
| Case1 | Max JIF (5 years) | 1.4353 | 1.3320–1.5561 | 0.67 | 0.0437 |
| | Document Count (All years) | 1.5792 | 1.3466–1.9241 | 0.54 | 0.0580 |
| | Funding amount | 1.5024 | 1.1463–1.7336 | 0.00 | 0.3223 |
| Case2 | h-index | 1.9205 | 1.7374–2.2709 | 1.00 | 0.0064 |
| | Max Cited By (All years) | 1.2006 | 1.1604–1.2390 | 0.00 | 0.3752 |
| | Funding amount | 1.5903 | 1.2180–1.9213 | 0.00 | 0.3153 |
| Case3 | h-index | 1.2381 | 0.9480–1.6305 | 0.00 | 0.5338 |
| | Funding amount | 4.7952 | 1.1918–14.0361 | 0.00 | 0.3110 |
| Case4 | Avg JIF (5 years) | 4.0141 | 3.4343–4.6601 | 1.00 | 0.0022 |
| | Funding amount | 1.5420 | 1.4851–1.6125 | 0.00 | 0.2460 |

* *sig_rate_0.05* denotes the proportion of repetitions (out of R = 100) in which the coefficient p-value was < 0.05.

technology field. This tendency likely reflects the generally higher impact levels of journals in biomedical research compared with Non-Biomedical areas. In Cases 1 and 2, where no field distinction was made, publication-quality variables showed weaker discriminative value. This may reflect cross-field heterogeneity in citation practices, suggesting that field-normalized indicators (e.g., field-normalized citation rates) could be more appropriate when modelling publication-related metrics in mixed-field samples. However, when the analysis was stratified by field—as in Cases 3 and 4—these indicators became more discriminative and effective as metrics. Based on these insights, JIF can be regarded as a representative indicator for defining approximate thresholds: ≥ 20 for the biomedical field and ≥ 10 for Non-Biomedical fields. The threshold for the biomedical field is derived from the results of Case 3, while the threshold of JIF ≥ 10 for Non-Biomedical fields reflects the original data scope of Case 4, which only included papers published in journals above that level.

In summary, key characteristics of researchers associated with the growth of university-originated startups include securing a cumulative Funding Amount of at least 10 million yen as a principal investigator, and/or being the author of papers with JIF ≥ 20 in the biomedical field or JIF ≥ 10 in the Non-Biomedical field. These characteristics suggest that researchers with highly regarded academic outputs are more likely to contribute to startup growth.

**Identification of researcher performance at universities that rank highly in the number of university-originated startups they produce in Japan**

Based on these characteristics, we examined the research performance of top six universities identified by the Ministry of Economy, Trade and Industry's "Survey on the Status of University-Based Ventures" [23]: The University of Tokyo, Keio University, Kyoto University, Osaka University, University of Tsukuba, and Tohoku University. **This university-level assessment is intended as a supplementary application of the identified characteristics and is conceptually separate from the policy-context description in Section 3, where four major national universities were mentioned as an illustrative example of the Public-Private Innovation Program. Accordingly, the mention of those four universities does not imply that the startup-level analytical sample was restricted to any specific subset of universities.**

Fig 6 (a) shows the distribution of cumulative KAKENHI grants (threshold: 10 million JPY) for our model dataset. Overall, about 25% of researchers exceeded 10 million JPY. In the biomedical and healthcare fields, roughly 30% passed the threshold. Among researchers from the top six universities, this proportion increased to around 40%.

Next, we extracted data from "KAKEN" for researchers at the top six universities who received Section I (Biomedical & Healthcare) awards in FY 2024. The results, shown in Fig 6 (b), indicate that approximately 70% of these researchers had received grant amounts exceeding the threshold, revealing a large gap compared to the model dataset.

Fig 7 (a) shows the distribution of JIF values (thresholds: 10 and 20) by technological domain for the dataset used in model construction. Overall, papers with JIF < 10 constituted the majority, particularly in AI & IoT, Robotics, and Electronics. In contrast, fields such as Biomedical and Environment & Energy had more papers with JIF ≥ 10. As a representative example from a Non-Biomedical domain, we extracted JIF values for researchers at the top six universities whose 2024 KAKENHI awards fell under Materials categories (Medium Category: 18–19 and 26–30). The lower panel of Fig 7 (b) indicates that, in this Materials subset, JIF ≥ 10 again accounted for the majority—demonstrating a distribution distinct from the model dataset.

## 6. Conclusion

This study aimed to develop an identification model that identifies researchers contributing to deep-tech startup growth. We used the Ministry of Economy, Trade and Industry's University-Originated Venture Database to build a machine learning classification model, where predictor variables included publication metrics and KAKENHI records, and target variable was startup funding amount. The model achieved an accuracy range of 67–81%, with higher performance observed when the distinction between groups was clearer—either by setting a larger funding threshold or by stratifying the data into biomedical and Non-Biomedical domains. Feature-importance and breakdown-tree analyses further revealed key researcher

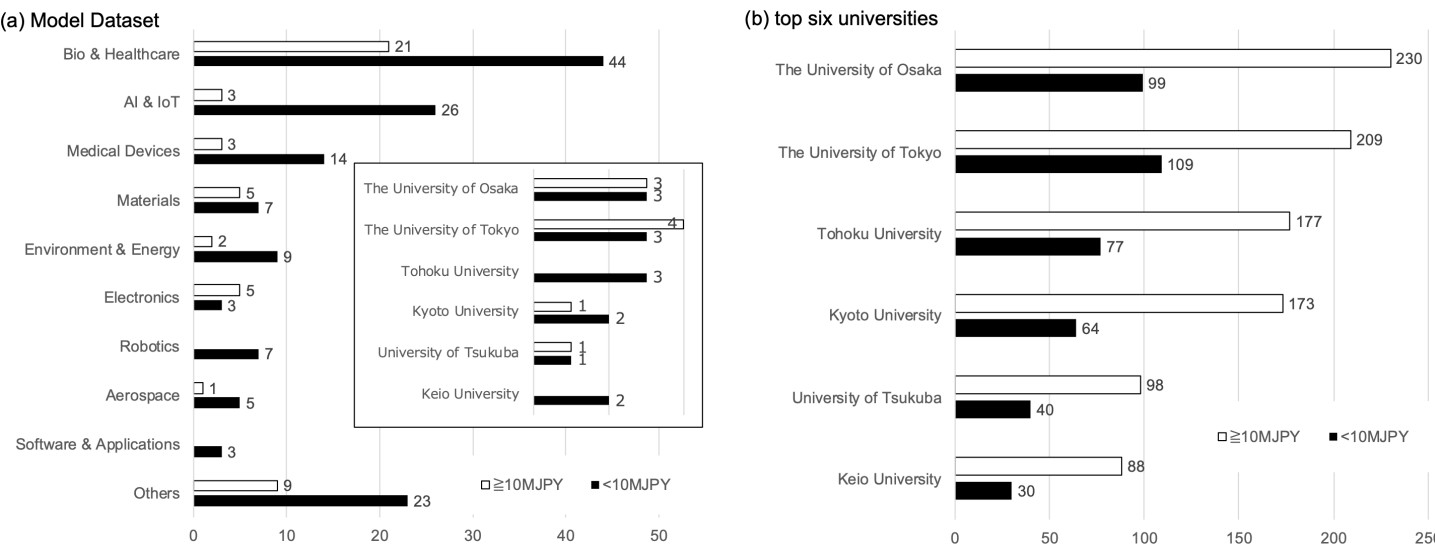

**Fig 6. KAKENHI grant amounts distribution.**

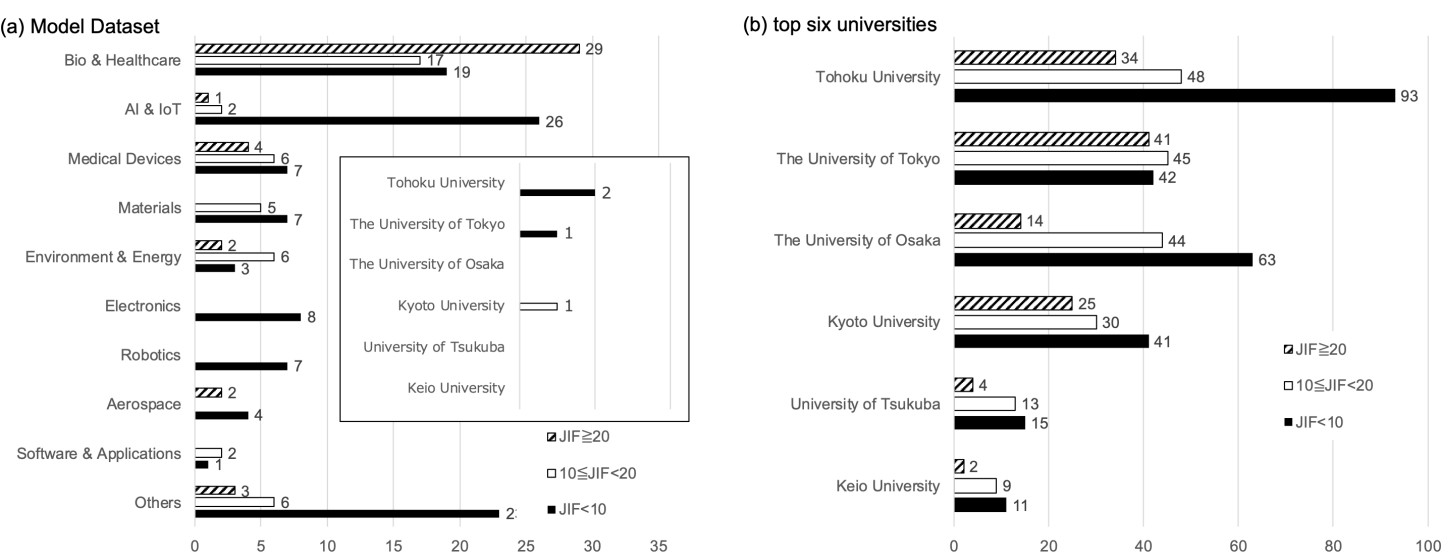

**Fig 7. JIF Distribution.**

characteristics associated with startup growth. To support interpretation, we additionally conducted logistic regression analyses and reported odds ratios with confidence intervals and p-values; these results are presented as complementary evidence of association rather than causal effects.

Specifically, the study identified two primary characteristics of such researchers: receiving a cumulative KAKENHI grant of at least 10 million yen as a principal investigator, and being an author of papers with a JIF of ≥20 in biomedical fields or ≥10 in Non-Biomedical fields. These characteristics were shown to be effective not only in reflecting the length of a research career but also in capturing the qualitative academic identification of the researchers' outputs.

Deep-tech startups require long-term R&D and large-scale investment, which inherently carry high risks. Therefore, establishing a methodology for qualitative identification of researchers at the early stage is crucial for enabling efficient and sustainable capital circulation, thereby maximizing the social and economic impact. It should also be acknowledged that our analyzed sample may not fully represent the entire population of Japanese university-affiliated startups, as the dataset was constructed under practical constraints of data availability and linkage feasibility across sources. Therefore, expanding, standardizing, and integrating relevant databases remain essential to strengthen the robustness and external validity of this line of research. At the same time, it is important to clarify that our analytical dataset should be regarded as a selected subset conditioned on successful cross-database linkage and the availability of researcher identifiers and related records, rather than as a bias-free random sample. These inclusion conditions could plausibly be associated with fundraising visibility and disclosure practices, and therefore the observed associations are best interpreted as patterns within this selected, investment-observable subset. In addition, for startups involving multiple researchers, our one-to-one data construction relied on repeated random selection (five iterations), which should be interpreted as a limited sensitivity check rather than a definitive assessment of model stability. Accordingly, generalization to the full population should be made with caution.

Furthermore, this study highlighted a persistent gap between the research capabilities of individual scholars and their actual involvement in startup formation. Even among researchers with strong publication and funding records, the translation of research excellence into entrepreneurial outcomes remains limited. This indicates that Japan's current mechanisms for linking academic achievement to startup creation are still insufficient. To bridge this gap, it will be necessary to develop more effective institutional frameworks—such as university-based innovation offices, translational research programs, and dedicated venture development platforms—that actively support researchers in converting their scientific outcomes into viable businesses.

In parallel, building comprehensive and interoperable databases that integrate information on research outputs, intellectual property, and startup performance will be indispensable. Such datasets will not only enable more robust empirical research but also provide evidence-based foundations for government policies, including the "Startup Development Five-Year Plan" and other initiatives promoting university–industry collaboration. Through systematic data accumulation and the establishment of feedback mechanisms between academia, investors, and policymakers, Japan can strengthen its capacity to generate globally competitive deep-tech startups.

The identification model and insights obtained through this study are expected to contribute to the development of a more robust decision-making foundation for deep-tech investment and to the design of future policies that effectively translate research excellence into entrepreneurial impact.

## Supporting information

**S1 File. Dataset used for linkage and analysis of university-originated startups and researcher information.** (XLSX)

## Author contributions

**Data curation:** Yoshifumi Mizuhara.

**Formal analysis:** Yoshifumi Mizuhara.

**Writing – original draft:** Yoshifumi Mizuhara.

**Writing – review & editing:** Tetsusei Kurashiki.

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
