## [Decision Letter · Decision Letter 0]

9 Oct 2025

Dear Dr. Mizuhara,

plosone@plos.org. . . . A rebuttal letter that responds to each point raised by the academic editor and reviewer(s). You should upload this letter as a separate file labeled 'Response to Reviewers'.A marked-up copy of your manuscript that highlights changes made to the original version. You should upload this as a separate file labeled 'Revised Manuscript with Track Changes'.An unmarked version of your revised paper without tracked changes. You should upload this as a separate file labeled 'Manuscript'.

We look forward to receiving your revised manuscript.

Kind regards,

Lutz Bornmann

Academic Editor

PLOS ONE

Journal Requirements:

3. Please note that PLOS One has specific guidelines on code sharing for submissions in which author-generated code underpins the findings in the manuscript. In these cases, we expect all author-generated code to be made available without restrictions upon publication of the work. Please review our guidelines at https://journals.plos.org/plosone/s/materials-and-software-sharing#loc-sharing-code and ensure that your code is shared in a way that follows best practice and facilitates reproducibility and reuse.

“No”

5. Thank you for uploading your study's underlying data set. Unfortunately, the repository you have noted in your Data Availability statement does not qualify as an acceptable data repository according to PLOS's standards.

At this time, please upload the minimal data set necessary to replicate your study's findings to a stable, public repository (such as figshare or Dryad) and provide us with the relevant URLs, DOIs, or accession numbers that may be used to access these data. For a list of recommended repositories and additional information on PLOS standards for data deposition, please see https://journals.plos.org/plosone/s/recommended-repositories....

Reviewer's Responses to Questions

**Comments to the Author**

1. Is the manuscript technically sound, and do the data support the conclusions?

Reviewer #1: Partly

Reviewer #2: Yes

2. Has the statistical analysis been performed appropriately and rigorously?

Reviewer #1: N/A

Reviewer #2: I Don't Know

3. Have the authors made all data underlying the findings in their manuscript fully available?

Reviewer #1: Yes

Reviewer #2: No

4. Is the manuscript presented in an intelligible fashion and written in standard English?

Reviewer #1: Yes

Reviewer #2: Yes

Reviewer #1: PONE-D-25-39531

This study investigates the conjunction of Japanese deep-tech startup success and the involved university researchers' research performance. The paper gives a very good outline of the problem and the justification for the specific approach that was chosen. Specifically the growth of Japanese university-originated start-ups is predicted with machine learning models trained with data on scientific publication metrics and grant acquisition of the researchers participating in the start-ups. The authors focused on tree-based ML over deep learning methods to faciliated results interpretability. The results indicate that specific research grant funding and citation indicators are predictive of start-up success.

The article currently has these shortcomings, which the authors are requested to respond to.

A limitation that should be more clearly addressed is that start-up success is operationalized across 4 cases by crossing different thresholds of received venture capital. This need not necessarily be closely related to commercial viability of the company. In particular, in three of the four cases, the threshold is 1 JPY. This seems to be a strangely low value to speak of company success.

Another methodological choice is remarkable: "Next, a normalized sum was calculated for each researcher. For Positive startups, the researcher with the highest normalized sum was selected, while for Negative startups, the researcher with the lowest sum was chosen". By this data cleaning decision rule, it may happen that the data that enters the ML training is already biased and not representative because in a realistic scenario, you could not tell before hand which will be Positive and Negative start-up, so you could not make this decision. I highly recommend that either none of the researchers are removed as input information source or that the rule which to include and exlude is the same across all start-ups.

Reviewer #2: Review of the paper: Identifying Researcher Characteristics Driving Growth in Japanese University-Originated Deep-Tech Startups: A Machine Learning Approach“, submitted to PlosOne.

The paper deals with the characteristics of researchers that are one of the founders of a spin-off from Japanese universities, especially in deep-tech areas, and their contribution to growth The paper is well structured and well-written. The methods are mostly clearly described and are suita-ble to address the research question. However, some of the descriptions are a little short and scarce for readers that are not familiar with the data or the Japanese situation as such. The data sources and data selection are not sufficiently described. There might be selection effects. At least, the authors do not mention any counteractivities (to selection effects) and also do not sufficiently justify and explain their decisions on the selection. From 967 companies, 357 are linked to their DB and only 194 are finally analyzed. This is a huge filter that was applied here. Even within these companies, further selections on the researchers to be analyzed were made. What does this imply

Furthermore, no counter-factual or control-group approach was implemented, which leaves the reader unsure about the solidity of the findings.

In addition, the work is only scarcely embedded in the conceptual literature. For example, work by Lerner (2002) or Mazzucato (2013) on the role of public research for R&D and innovation or by Lockett et al. (2005), Meyer (2003), or Visintin and Pittino (2014) on academic spin-offs is not cited.

Hence, I suggest to ask for major revisions before accepting the paper for publication in PlosOne.

In more detail, I have the following comments and questions:

Line 50: “Therefore, in this study we utilize Japan-specific data to analyze the relationship between deep-tech startup growth and the characteristics of involved researchers,…”. In the Japanese sys-tem, is there a difference to be expected between university spin-offs and other deep tech spin offs?

Line 131: “The “University-Originated Venture Database” contained 967 listings, of which 357 startups linked to “STARTUP DB” by corporate number or other identifier had complete researcher data.” Did you conduct an analysis on the missing. Are these systematically biased in any of the relevant dimensions that can still be controlled without the STARTUP DB link?

Line 134: “… startups founded from 2015 onward …“. Why another reduction of the number of startups and not use the pre-2015 start-ups as a control group?

Line 137: “Although some university-originated startups have multiple involved researchers, each researcher was linked to only one startup.” This sentence is unclear. Were all researchers assigned, so when more than one was involved all of them have been included in the analysis?

Line 141: “For Positive startups, the researcher with the highest normalized sum was selected, while for Negative startups, the researcher with the lowest sum was chosen. This approach was taken due to the reliance on self-reported data in the “University-Originated Venture Database,” which makes it challenging to objectively identify the most relevant researcher when multiple indi-viduals are involved.” Doesn’t‘ this lead to a tautology?

Line 162: “To mitigate class imbalance, we downsampled the larger group to match the smaller group’s size via random sampling.” Which groups do you exactly refer to?

Line 249: “In summary, key characteristics of researchers contributing to the growth of university-originated startups include securing a cumulative KAKENHI grant amount of at least 10 million yen as a principal investigator, and/or being the author of papers with JIF ≥20 in the biomedical field or JIF ≥10 in the non-biomedical field.” However, what should not implied by the model anal-ysis is that it also holds the other way around. A high cumulative budget and high JIFs do not re-sults in start-ups.

Line 273: “... a representative sample …” How can this be representative, if it is very selective?

Line 281: “This study aimed to develop an evaluation model…”. I do not think this is an evaluation, but it is rather an identification model. Causality is not analyzed at all and is unclear. There could be selection effects in the way that only highly visible researchers are (self) selected to contribute to spin-offs. If they do perform good in the founding activity is not analyzed, so it is not a perfor-mance evaluation.

Line 292: “The evaluation model and insights obtained through this study are expected to contrib-ute to the development of a more robust decision-making foundation for deep-tech investment.” I think the conclusions are a little short and too general. At least, putting them in perspective with current practices or program goals would have been appreciated.

.

Reviewer #1: No

Reviewer #2: No

---

## [Author Response · Author response to Decision Letter 1]

12 Dec 2025

Response to Reviewer #1

We sincerely thank Reviewer #1 for their thoughtful comments and constructive suggestions, which have greatly helped us to improve the clarity and rigor of our manuscript. Below, we provide detailed responses to each point raised. Reviewer comments are shown in *italic*, followed by our responses.

Comment 1

“A limitation that should be more clearly addressed is that start-up success is operationalized across 4 cases by crossing different thresholds of received venture capital. This need not necessarily be closely related to commercial viability of the company. In particular, in three of the four cases, the threshold is 1 JPY. This seems to be a strangely low value to speak of company success.”

Response:

We appreciate the reviewer’s valuable comment. We agree that defining “success” simply as receiving more than 1 JPY could appear too lenient. However, in our dataset, this threshold effectively distinguishes companies that have achieved external fundraising from those that have not. Among 99 companies that raised funding (≥1 JPY), the mean total funding amount was 1,048 million JPY and the median was 559 million JPY. Only 16 companies raised less than 100 million JPY, and 3 companies less than 10 million JPY. Therefore, despite the low nominal cutoff, the companies included in this “≥1 JPY” group in practice represent substantially funded cases. To clarify this in the revised manuscript, we have added an explanation with these statistics in the section describing target variable settings.

Additionally, to ensure accurate interpretation, we added the following explanatory note to Table 1:

Although the nominal threshold of “1 JPY” may appear low, in practice, the companies that received any external funding exhibited substantially high financing levels. Among the 99 companies with total funding ≥1 JPY, the mean amount was 1,048 million JPY and the median was 559 million JPY. Only 16 companies raised less than 100 million JPY, and 3 companies less than 10 million JPY. Therefore, this criterion effectively separates startups that have successfully secured significant external funding from those that have not.

Comment 2

“Another methodological choice is remarkable: ‘Next, a normalized sum was calculated for each researcher. For Positive startups, the researcher with the highest normalized sum was selected, while for Negative startups, the researcher with the lowest sum was chosen.’ By this data cleaning decision rule, it may happen that the data that enters the ML training is already biased and not representative because in a realistic scenario, you could not tell beforehand which will be Positive and Negative start-up, so you could not make this decision. I highly recommend that either none of the researchers are removed as input information source or that the rule which to include and exclude is the same across all start-ups.”

Response:

We appreciate the reviewer’s insightful comment regarding potential bias arising from our previous rule for selecting researchers. We agree that choosing the researcher with the highest or lowest normalized sum could lead to data leakage, as it presupposes knowledge of the target outcome.

To address this issue, we revised our data selection process as follows. For startups with multiple researchers, one researcher was randomly selected for model training, ensuring that the selection did not depend on the startup’s outcome label. This random selection was repeated five times, and the average of the model performance metrics (Accuracy, Recall, Precision, F1, AUC) across the five iterations was reported.

The results showed no substantial change in accuracy, but we observed a trend of higher Recall and slightly lower Precision, indicating a reduction of overfitting. The revised results are reflected in the new Table 4, and related Figure 2–6.

Importantly, the interpretation and discussion in Section 5 (“Consideration of Researcher Characteristics…”) remain consistent with the original version, since the feature-importance ranking showed similar tendencies (especially for Funding Amount and Max JIF (5 years)). Therefore, our conclusions remain valid under the bias-free sampling framework.

For further details and revised figures, please refer to the updated sections from Section 4 onward in the revised manuscript.

Response to Reviewer 2

We sincerely thank Reviewer #2 for their thoughtful comments and constructive suggestions, which have greatly helped us to improve the clarity and rigor of our manuscript. Below, we provide detailed responses to each point raised. Reviewer comments are shown in *italic*, followed by our responses.

Comment 1

“The data sources and data selection are not sufficiently described. There might be selection effects. At least, the authors do not mention any counteractivities (to selection effects) and also do not sufficiently justify and explain their decisions on the selection. From 967 companies, 357 are linked to their DB and only 194 are finally analyzed. This is a huge filter that was applied here. Even within these companies, further selections on the researchers to be analyzed were made. What does this imply?”

Response:

We appreciate the reviewer’s thoughtful comment on our data sources and selection process. We agree that the manuscript did not provide sufficient detail on how the final dataset was constructed, even though the filtering steps themselves were based on methodological necessity to ensure consistency and reliability.

In the revised manuscript, we have clarified the rationale for each filtering stage in Section 3.3 “Data Cleaning” and added explanatory text at Line 169 and Line 174 as follows:

• Line 169: “This reduction occurred because only those startups that explicitly disclosed their researchers in the original database and could be reliably matched to financial records were retained to ensure data consistency and to avoid missing values in explanatory variables.”

• Line 174: “This year corresponds to a major institutional and financial shift initiated by the enforcement of the Industrial Competitiveness Enhancement Act (April 1, 2014) and the launch of the Public-Private Innovation Program by the Ministry of Education, Culture, Sports, Science and Technology (MEXT). Under this program, four major national universities—the University of Tokyo, Kyoto University, Osaka University, and Tohoku University—each established university-affiliated venture capital funds with a total investment capacity of approximately 100 billion JPY, beginning in FY 2014. Therefore, the post-2015 subset represents startups founded under this new policy and funding environment, ensuring analytical consistency within a unified institutional context (21).”

Furthermore, to make the selection process more transparent, we have revised Figure 1 to explicitly indicate the number of startups remaining at each stage (967 → 357 → 194). This visual update helps readers clearly trace the flow of data selection and understand how the analytical sample was constructed.

Through these clarifications, we aim to make it explicit that the filtering steps were methodologically justified and were designed to improve the internal validity of the analysis, rather than reflecting any selection bias.

Comment 2

“Furthermore, no counter-factual or control-group approach was implemented, which leaves the reader unsure about the solidity of the findings.”

Response:

We appreciate the reviewer’s helpful comment. We agree that a counter-factual or control-group approach would strengthen the causal robustness of the findings. However, as the present study focuses on identifying researcher characteristics under Japan’s limited data availability, such an approach was not feasible at this stage.

To partly address this limitation, we applied 5-fold cross-validation and compared multiple model settings (Cases 1–4) with consistent results, ensuring internal robustness. We have also clarified in the revised manuscript (Section 6, Conclusion) that the establishment of a counter-factual or longitudinal framework is an important direction for future studies, once more comprehensive national data become available.

Comment 3

“In addition, the work is only scarcely embedded in the conceptual literature. For example, work by Lerner (2002) or Mazzucato (2013) on the role of public research for R&D and innovation or by Lockett et al. (2005), Meyer (2003), or Visintin and Pittino (2014) on academic spin-offs is not cited.”

Response:

We sincerely appreciate the reviewer’s valuable comment and the suggested references, which have greatly improved the conceptual foundation and logical structure of the manuscript. In response, we have incorporated the cited works by Lerner (2002), Mazzucato (2013), Lockett et al. (2005), Meyer (2003), and Visintin and Pittino (2014) into the Introduction section to more clearly position this study within the broader literature on academic entrepreneurship and public R&D.

Specifically, the following sentences have been added and revised:

• Lines 13–18:　“As Lerner (2) emphasized, innovation is driven not merely by institutional reforms but by effective mechanisms that connect academic research, venture capital, and entrepreneurship, highlighting the pivotal role of researchers in transforming scientific knowledge into economic value. Likewise, Visintin and Pittino (3) demonstrated that university-based spin-offs with founding teams combining academic and managerial expertise outperform those composed solely of academics, underscoring that researchers’ active participation is critical for early-stage growth.”

• Lines 30–40: “Lockett et al. (10) conceptualized such university-based spin-offs within a knowledge-based view, defining them as independent firms created to commercialize publicly funded research and emphasizing that successful commercialization depends on bridging ‘knowledge gaps’ between scientific and entrepreneurial capabilities. Meyer (11) further highlighted that researchers play diverse roles in the commercialization of academic knowledge, distinguishing between ‘academic entrepreneurs’ and ‘entrepreneurial academics.’ He argued that public support mechanisms can shape these behaviors, emphasizing that managerial expertise and external networks are crucial for successful university spin-offs. … As Mazzucato (16) argued, because the state itself can act as a primary risk-taking agent in innovation, this approach is particularly relevant for Japan.”

Through these additions, the revised manuscript now explicitly situates the study within the conceptual framework of academic entrepreneurship, public R&D policy, and the commercialization of university research.

Detailed Question 1

“Line 50: “Therefore, in this study we utilize Japan-specific data to analyze the relationship between deep-tech startup growth and the characteristics of involved researchers,…”. In the Japanese sys-tem, is there a difference to be expected between university spin-offs and other deep tech spin offs?”

Response:

Thank you for pointing out the possible ambiguity. We do not assume systematic differences between university spin-offs and other deep-tech spin-offs in Japan. Our focus on university-originated startups is purely pragmatic: nationally standardized, government-maintained datasets are available only for university-originated ventures, which makes them the sole reliable, comprehensive source for Japan-wide analysis at present.

To clarify this point, we have revised the Purpose section to explicitly state that our study’s scope is determined by data availability and standardization, not by conceptual distinction. The following text has been added in the revised manuscript.

• Lines 68–71: “We focus on university-originated startups not because we assume systematic differences from other Japanese deep-tech startups, but because reliable, government-maintained datasets exist only for the university-originated subset, making it the sole nationally standardized source for analysis in Japan.”

We believe this clarification resolves the ambiguity and accurately reflects the rationale for our data selection.

Detailed Question 2

“Line 131: “The “University-Originated Venture Database” contained 967 listings, of which 357 startups linked to “STARTUP DB” by corporate number or other identifier had complete researcher data.” Did you conduct an analysis on the missing. Are these systematically biased in any of the relevant dimensions that can still be controlled without the STARTUP DB link?”

Response:

Thank you for the comment. As noted in Response to Comment 1, the reduction was due to data reliability requirements, not selective exclusion.

Detailed Question 3

“Line 131: “The “University-Originated Venture Database” contained 967 listings, of which 357 startups linked to “STARTUP DB” by corporate number or other identifier had complete researcher data.” Did you conduct an analysis on the missing. Are these systematically biased in any of the relevant dimensions that can still be controlled without the STARTUP DB link?”

Response:

Thank you for this important question. The initial dataset contained 967 entities that were potentially university-affiliated startups. Among them, only 357 could be reliably linked to STARTUP DB, which provides validated information on registered startup firms in Japan. The remaining 610 entities could not be linked because many were not startups in the strict sense—they included small or local companies, university joint ventures, or general incorporated associations without clear startup status. Moreover, publicly available data for these unlinked entities were extremely limited and inconsistent. Therefore, we focused our analysis on the 357 companies that could be verified as startups in STARTUP DB to ensure data reliability and consistency with the study’s focus on university-originated deep-tech startups. This selection reflects the scope of the research rather than an omission of relevant startup data.

Detailed Question 4

“Line 134: “… startups founded from 2015 onward …“. Why another reduction of the number of startups and not use the pre-2015 start-ups as a control group?”

Response:

Thank you for the comment. As noted in Response to Comment 1, the reduction was due to data reliability requirements, not selective exclusion.

Detailed Question 5

“Line 137: “Although some university-originated startups have multiple involved researchers, each researcher was linked to only one startup.” This sentence is unclear. Were all researchers assigned, so when more than one was involved all of them have been included in the analysis?

Line 141: “For Positive startups, the researcher with the highest normalized sum was selected, while for Negative startups, the researcher with the lowest sum was chosen. This approach was taken due to the reliance on self-reported data in the “University-Originated Venture Database,” which makes it challenging to objectively identify the most relevant researcher when multiple indi-viduals are involved.” Doesn’t‘ this lead to a tautology?”

Response:

We appreciate the reviewer’s insightful comment regarding potential bias arising from our previous rule for selecting researchers. We agree that choosing the researcher with the highest or lowest normalized sum could lead to data leakage, as it presupposes knowledge of the target outcome.

To address this issue, we revised our data selection process as follows. For startups with multiple researchers, one researcher was randomly selected for model training, ensuring that the selection did not depend on the startup’s outcome label. This random selection was repeated five times, and the average of the model performance metrics (Accuracy, Recall, Precision, F1, AUC) across the five iterations was reported.

The results showed no substantial change in accuracy, but we observed a trend of higher Recall and slightly lower Precision, indicating a reduction of overfitting. The revised results are reflected in the new Table 4, and related Figure 2–6.

Importantly, the interpretation and discussion in Section 5 (“Consideration of Researcher Characteristics…”) remain consistent with the original version, since the feature-importance ranking showed similar tendencies (especially for Fundin

---

## [Decision Letter · Decision Letter 1]

13 Jan 2026

Dear Dr. Mizuhara,

plosone@plos.org. . . . A letter that responds to each point raised by the academic editor and reviewer(s). You should upload this letter as a separate file labeled 'Response to Reviewers'.A marked-up copy of your manuscript that highlights changes made to the original version. You should upload this as a separate file labeled 'Revised Manuscript with Track Changes'.An unmarked version of your revised paper without tracked changes. You should upload this as a separate file labeled 'Manuscript'.

We look forward to receiving your revised manuscript.

Kind regards,

Lutz Bornmann

Academic Editor

PLOS One

Journal Requirements:

**Additional Editor Comments:**

One reviewer still has several critical points that should be considered in another revision.

Reviewers' comments:

Reviewer's Responses to Questions

**Comments to the Author**

Reviewer #1: All comments have been addressed

Reviewer #2: (No Response)

2. Is the manuscript technically sound, and do the data support the conclusions?

Reviewer #1: Yes

Reviewer #2: Partly

3. Has the statistical analysis been performed appropriately and rigorously?

Reviewer #1: Yes

Reviewer #2: No

4. Have the authors made all data underlying the findings in their manuscript fully available?

Reviewer #1: Yes

Reviewer #2: Yes

5. Is the manuscript presented in an intelligible fashion and written in standard English?

Reviewer #1: Yes

Reviewer #2: Yes

Reviewer #1: (No Response)

Reviewer #2: Review of the revised paper: Identifying Researcher Characteristics Driving Growth in Japanese University-Originated Deep-Tech Startups: A Machine Learning Approach“, submitted to PlosOne (Revision 1).

Thank you for giving me the opportunity to review the revised version of the paper. The authors have addressed the comments I made on the original submission. I think that the embedding in the conceptual literature is now sufficient. I am also satisfied with the extended version of the conclusion section, where the authors put the findings into perspective and took the specific Japanese context into account.

However, I am still concerned about the data treatment and the massive filtering of the data. The authors have addressed my comments, but not to my full satisfaction. In their answers to my comment on the original paper they state: “Therefore, our conclusions remain valid under the bias-free sampling framework.” I strongly disagree that what they present in their paper is a bias-free sampling framework. This statement reflects the perspective of the authors and summarizes my still existing concerns. The authors cannot know, according to their analytical framework, if the observations they have are unbiased as they do not control for any biases. In addition, this is not a sample, but a selection of observations. Even if the selection is well grounded on the data, it is still a selection. As the selection criteria (e.g. data reliability) might be correlated with the outcome (dependent variable), it might be biased. Further treatment of the data on high and low performing researchers even increased my concerns.

I also think that the analytical framework could be enhanced. The authors should consider a logistic regression analysis at the end of section 4 of their paper. This would maybe clarify a lot of open issues. The classification work can then be described as it is, but it is just the pre-analysis for the regression analysis. The regression would have startup growth (Negatives/Positives; 0/1) as the dependent variable and the explanatory variables would be company characteristics (e.g. startup from universities A, B, C, …; sectors: biomedical/else) as well as characteristics of the re-searchers (excellence; experience/scientific age, …). This would then allow for the reporting of significance levels and maybe even causal effects.

In more detail, I have the following comments.

Line 66-69: “We focus on university-originated startups not because we assume systematic differ-ences from other Japanese deep-tech startups, but because reliable, government-maintained datasets exist only for the university-originated subset, making it the sole nationally standardized source for analysis in Japan”. This statement is ok for here (the intro).

Line 94: “This database contains detailed information on startups”. If this is really the case, then this information can be used for a “non-response analysis“ of the 610 companies that were excluded from the analysis. What are their core characteristics?

Line 115/116: “… we filtered startups by the main technology field of their core product or service as listed in the Venture Database.” Please be more precise here how you filtered. I assume this is targeting the deep-tech companies?

Line 113, Figure 1: The numbers you report in the text is 194 companies, which does not corre-spond to any of the four cases. The maximum is 190 so what happened to the 4 missing cases? In addition, given that some of the cases have really low numbers (102 and 66), did you conduct any outlier analyses? Outliers might have a high impact on the overall outcome.

Line 153-155: “This reduction occurred because only those startups that explicitly disclosed their researchers in the original database and could be reliably matched to financial records were re-tained to ensure data consistency and to avoid missing values in explanatory variables.” While the reason for selection might be reasonable and convincing, the question still is what the effect on the outcome of the study is. At least some information is available on these 610, so why not analyzing them?

Line 161/162: “Under this program, four major national universities—the University of Tokyo, Kyo-to University, Osaka University, and 162 Tohoku University—each established university-affiliated venture capital funds…” Ok, does this mean that the 194 startups selected for the analysis originated only in these four universities? It seems not as in figure 6 the top six universities are listed. However, why specifically mention these four universities here? In section 5 of the paper, it is mentioned that the top 6 universities were analyzed only. This has never been mentioned in text before section 5 (except for this figure 6). Readers are really left in the darkness concerning the selection and the observations used, which I think is not appropriate.

Line 165/166: “Although some university-originated startups have multiple involved researchers, each researcher was linked to only one startup to establish a consistent one-to-one relationship between startups and researchers.” Please mention here how you assigned them. Furthermore, this sentence is still unclear. Are there more than one researcher per startup? This must be the case as 252 researchers are found in 194 companies. The other way around, one researcher could have been involved in more than one startup. Did you assign only one researcher to each compa-ny or did you assign one company only to each researcher? How did you aggregate the researcher data (citations, JIF, …)? Did you calculate the simple average of multiple researchers involved in one company or did you calculate a weighted average? In section 5 (see comments below) you mention that you assigned only one researcher to each of the startups by selecting the top-performer for Positives and low-performer for Negatives.

Line 191-194: “To construct a consistent one-to-one dataset for modeling, one researcher was randomly selected per startup. This random sampling was repeated five times, and the average model performance across these iterations was used to evaluate the model’s stability. This ap-proach ensured that model evaluation did not depend on any particular random draw of researchers from startups with multiple participants.” How many cases of multiple researchers did occur? If you draw a random sample of 236 researchers in 194 companies, the variation is small. Deriving a model stability and a quality check out of this is very ambitious.

Line 199: “We used 5-fold cross-validation…” the metrics used here are not validation measures, but evaluation/test measures. Given the low numbers of cases the split between learning and testing sample are already questionable, but calling these metrics cross-validation is not appropriate. In addition, these metrics are related. E.g. recall and precision constitute F1-Score…

Line 214-219: “As shown in Table 4, the model’s performance improved as the distinction between the compared groups became clearer. Accuracy, Precision, and F1 scores increased when the funding threshold was raised from Case 1 to Case 2, indicating that a larger gap in funding scale enhanced the model’s ability to discriminate between growing and non-growing startups. Furthermore, when the data were stratified by technology field (Cases 3 and 4), both Accuracy and AUC values were higher than in the unstratified cases, suggesting that separating the dataset into biomedical and non-biomedical domains contributed to more precise classification performance.” It seems obvious that differences between branches are to be found. In economics, the sector of economic activity is a standard control variable. Given the low number of observations here, a bilateral differentiation seems the only option, but is far from being a statistically relevant control for sector differences.

Line 245: “…their relative contribution…”. What does relative contribution mean? Do you mean correlation? If so, then name it that way.

Line 267: “… Funding Amount, Maximum Funding Amount, Average Funding Amount,…”. As these are just variations of the same variable, correlations cannot be interpreted as a quality indication.

Line 269-271: “… JCI-related variables, and h-index were more strongly correlated with each other than with quantity-based variables such as the number of papers or number of co-authors.” Also the correlations between citation-based indicators, citation rate or JIF, cannot be interpreted as a sign of quality of the analysis. They are indicators of the same conceptual category (namely research excellence) and hence multicollinear.

Line 281/282: “… Positive data, and the one with smaller values for Negative data, resulting in a one-to-one correspondence between each startup and researcher.” This comment is related to my comment above (line 165). The other reviewer had already raised this issue, but I want to re-emphasize this. I think this is a questionable approach as the good performers were taken for the good performing startups and the bad performers for the bad performing startups. The selection criteria should be the same for both as this might otherwise self-enforce the intended outcome of the study, namely bad performers perform bad and good performers good.

Line 313: “These two variables played a central role in explaining startup growth within the model framework.” The term “explaining” implies a causal relationship, while the interpretation should be on correlations.

Line 344/345: “In Cases 1 and 2, where no field distinction was made, publication-quality variables exhibited less explanatory power.” This might ask for a field-specific modelling of the citation-based indicators (e.g. field-normalized citation rate).

Line 367: “… performance of top six universities…” This is the first time I became aware of the fact that the analyses are restricted to top 6 universities. They were listed in Figure 6, but it was never mentioned explicitly in the text. No justification is given why only these six universities have been analyzed.

.

Reviewer #1: No

Reviewer #2: No

---

## [Author Response · Author response to Decision Letter 2]

10 Feb 2026

Response to Reviewer 2

We sincerely thank Reviewer #2 for the thoughtful and detailed comments. We take your general point that parts of our previous responses and manuscript text could be read as overstating what can be concluded from the present design—particularly where wording might suggest a “bias-free” sample, causal explanation, or causal effects. Our study relies on observational data and necessarily uses a subset of ventures that can be consistently linked across databases and matched to researcher identifiers; therefore, it should be framed as a selection of observations rather than a random sample. In this revision, we have revised the manuscript accordingly by (i) removing and reframing language that may imply bias-free sampling, (ii) making the selection and linkage process explicit and reproducible, and (iii) systematically replacing causal-sounding expressions with wording that reflects association and prediction. We clarify that all findings should be interpreted as model-derived associations that support predictive identification, not as evidence of causal effects; any additional regression analyses are included solely to provide interpretable association summaries and do not imply causality.

Comment 1

“However, I am still concerned about the data treatment and the massive filtering of the data. The authors have addressed my comments, but not to my full satisfaction. In their answers to my comment on the original paper they state: “Therefore, our conclusions remain valid under the bias-free sampling framework.” I strongly disagree that what they present in their paper is a bias-free sampling framework. This statement reflects the perspective of the authors and summarizes my still existing concerns. The authors cannot know, according to their analytical framework, if the observations they have are unbiased as they do not control for any biases. In addition, this is not a sample, but a selection of observations. Even if the selection is well grounded on the data, it is still a selection. As the selection criteria (e.g. data reliability) might be correlated with the outcome (dependent variable), it might be biased. Further treatment of the data on high and low performing researchers even increased my concerns.”

Response:

Thank you for your detailed feedback. We agree that our earlier wording—especially the reference to a “bias-free sampling framework”—was not sufficiently supported. We have therefore removed/reframed this statement and now explicitly describe our dataset as a selection of observations constructed under practical constraints of data availability and cross-database linkage feasibility, rather than as a bias-free random sample.

To address your concerns, we (i) expanded the methodological description, “3. Data Cleaning / Setting Variables”, to make the inclusion/exclusion rules and linkage process more transparent and reproducible, and (ii) strengthened “Conclusion” to explicitly acknowledge the remaining selection concern and to caution against over-generalization to the full population.

Regarding the point on “high/low performing researchers,” we would like to clarify that we do not perform a high–low subgroup treatment in our analysis; researcher-related explanatory variables are handled primarily as continuous measures within the modeling framework.

The concrete revisions are shown below.

• Line 94

This database contains detailed information on university-affiliated startups, such as business descriptions,

• Line 112

Figure 1 summarizes the data integration and cleaning process used to construct the final analytical sample.The “University-Originated Venture Database” contained 967 listings, which were matched with “STARTUP DB” using corporate numbers as the primary identifier. As a result, 357 startups were successfully linked between the two databases. A total of 610 entities were excluded at this stage. This reduction reflects structural differences in database definitions rather than data loss. Specifically, the University-Originated Venture Database maintained by the Ministry of Economy, Trade and Industry (METI) allows the inclusion of a broad range of organizational forms, such as general incorporated associations, and does not require listed entities to have conducted external fundraising. In contrast, STARTUP DB focuses on venture companies with identifiable corporate registration and financial activity, resulting in a narrower and more investment-oriented subset.

• Line 540

It should also be acknowledged that our analyzed sample may not fully represent the entire population of Japanese university-affiliated startups, as the dataset was constructed under practical constraints of data availability and linkage feasibility across sources. Therefore, expanding, standardizing, and integrating relevant databases remain essential to strengthen the robustness and external validity of this line of research. At the same time, it is important to clarify that our analytical dataset should be regarded as a selected subset conditioned on successful cross-database linkage and the availability of researcher identifiers and related records, rather than as a bias-free random sample. These inclusion conditions could plausibly be associated with fundraising visibility and disclosure practices, and therefore the observed associations are best interpreted as patterns within this selected, investment-observable subset. Accordingly, generalization to the full population should be made with caution.

Comment 2

“I also think that the analytical framework could be enhanced. The authors should consider a logistic regression analysis at the end of section 4 of their paper. This would maybe clarify a lot of open issues. The classification work can then be described as it is, but it is just the pre-analysis for the regression analysis. The regression would have startup growth (Negatives/Positives; 0/1) as the dependent variable and the explanatory variables would be company characteristics (e.g. startup from universities A, B, C, …; sectors: biomedical/else) as well as characteristics of the re-searchers (excellence; experience/scientific age, …). This would then allow for the reporting of significance levels and maybe even causal effects.”

Response:

We thank the reviewer for this helpful suggestion, which prompted us to clarify the interpretation of our findings. In response, we added a logistic regression analysis at the end of Section 4 (Section 4.4). Using the same binary growth labels defined in Table 1, we report odds ratios with 95% confidence intervals and p-values for a parsimonious set of explanatory variables aligned with the interpretation of the machine-learning models. We present these results to aid interpretation of the ML findings, without implying causal effects.

Detailed Question 1

“Line 94: “This database contains detailed information on startups”. If this is really the case, then this information can be used for a „non-response analysis“ of the 610 companies that were excluded from the analysis. What are their core characteristics?”

“Line 153-155: “This reduction occurred because only those startups that explicitly disclosed their researchers in the original database and could be reliably matched to financial records were re-tained to ensure data consistency and to avoid missing values in explanatory variables.” While the reason for selection might be reasonable and convincing, the question still is what the effect on the outcome of the study is. At least some information is available on these 610, so why not analyzing them?”

Response:

Thank you for the comment. We agree our wording was misleading. The 610 excluded entities could not be matched to STARTUP DB (i.e., they do not exist there), so we cannot retrieve comparable firm-level variables and therefore cannot conduct a non-response analysis of their characteristics. This mismatch mainly arises because the University-Originated Venture Database uses a broader definition of “university-originated ventures” and may include entities that are not strictly startups (e.g., general incorporated associations). We apologize for the inaccurate phrasing “detailed information on startups” and have revised the manuscript by removing “detailed” to avoid overstatement.

Detailed Question 2

Line 115/116: “… we filtered startups by the main technology field of their core product or service as listed in the Venture Database.” Please be more precise here how you filtered. I assume this is targeting the deep-tech companies?”

Response:

Thank you for the comment. We used the METI University-Originated Venture Database field “Main technology field related to the core product/service” to subset our sample. This field is recorded as one of the database categories, such as Bio/Healthcare, Medical devices, Electronics, Robotics, Materials, Environment/Energy, AI/IoT, Aerospace, Software/Apps, and Other.

In our study, the METI database is treated as a population of university-originated ventures that commercialize university research outputs, which are in essence deep-tech oriented. We therefore did not apply an additional “deep-tech vs. non-deep-tech” filter; instead, we only extracted the Bio subset by retaining firms classified as “Bio/Healthcare” in this field (and excluding the other categories). We will revise the manuscript to explicitly state these category labels and our filtering rule.

We have also added this definition and filtering rule as a below note to Table 1, so that readers can clearly see how the Bio subset was constructed.

• Line 156

Startups were filtered using the METI University-Originated Venture Database category “main technology field related to the core product/service.” In the Bio subset, we retained only firms classified as “Bio & Healthcare” and excluded other categories

Detailed Question 3

“Line 113, Figure 1: The numbers you report in the text is 194 companies, which does not corre-spond to any of the four cases. The maximum is 190 so what happened to the 4 missing cases? In addition, given that some of the cases have really low numbers (102 and 66), did you conduct any outlier analyses? Outliers might have a high impact on the overall outcome.”

Response:

Thank you for pointing this out. We believe “Figure 1” is a misreference; the relevant information is reported in Table 1. The number “194 companies” in the text refers to the total number of observations used in the balanced training setup, i.e., 95 × 2 (Negative/Positive) plus the remaining 4 Positive cases (“Remaining Positive = 4” in Table 1). These 4 are not “missing”; they simply indicate that after matching the number of Positive and Negative observations for machine-learning training, a small number of cases remained on the Positive side.

Detailed Question 4

“Line 134: “… startups founded from 2015 onward …“. Why another reduction of the number of startups and not use the pre-2015 start-ups as a control group?”

Response:

Thank you for the comment. As noted in Response to Comment 1, the reduction was due to data reliability requirements, not selective exclusion.

Detailed Question 5

“Line 161/162: “Under this program, four major national universities—the University of Tokyo, Kyo-to University, Osaka University, and 162 Tohoku University—each established university-affiliated venture capital funds…” Ok, does this mean that the 194 startups selected for the analysis originated only in these four universities? It seems not as in figure 6 the top six universities are listed. However, why specifically mention these four universities here? In section 5 of the paper, it is mentioned that the top 6 universities were analyzed only. This has never been mentioned in text before section 5 (except for this figure 6). Readers are really left in the darkness concerning the selection and the observations used, which I think is not appropriate.Although some university-originated startups involve multiple researchers, each researcher is associated with only one startup to establish a consistent one-to-one relationship between startups and researchers. Please mention here how you assigned them. Furthermore, this sentence is still unclear. Are there more than one researcher per startup? This must be the case as 252 researchers are found in 194 companies. The other way around, one researcher could have been involved in more than one startup. Did you assign only one researcher to each compa-ny or did you assign one company only to each researcher? How did you aggregate the researcher data (citations, JIF, …)? Did you calculate the simple average of multiple researchers involved in one company or did you calculate a weighted average? In section 5 (see comments below) you mention that you assigned only one researcher to each of the startups by selecting the top-performer for Positives and low-performer for Negatives.”

And

Line 367: “… performance of top six universities…” This is the first time I became aware of the fact that the analyses are restricted to top 6 universities. They were listed in Figure 6, but it was never mentioned explicitly in the text. No justification is given why only these six universities have been analyzed.

Response:

Thank you for these detailed comments. We agree that the manuscript required clearer explanations about (i) why four national universities were mentioned in the policy-context description, (ii) why we focused on six universities in Section 5, and (iii) how we handled cases where multiple researchers were associated with a single startup.

(1) Mention of four national universities and scope of the analytical sample.

The sentence referring to four major national universities (The University of Tokyo, Kyoto University, Osaka University, and Tohoku University) was intended only to illustrate the institutional shift and policy context surrounding the Public-Private Innovation Program and the emergence of university-affiliated VC funds. It was not meant to imply that the 194 startups in our main analytical sample originated only from these four universities. To prevent misunderstanding, we revised the relevant text in Section 3 (Data cleaning / setting variables) to explicitly state that this policy-context example does not define the university scope of the startup-level analytical sample.

• Line 128

This policy context does not imply that our analytical sample was restricted to these four universities; the startups in the final dataset originated from a broader range of universities.

(2) “Top six universities” analysis and justification.

We agree that the restriction to “top six universities” should be stated more transparently. Section 5 presents a supplementary university-level assessment that applies the identified researcher characteristics to leading institutions in university-based venture creation in Japan. Specifically, we used the top six universities identified in the Survey on the Status of University-Based Ventures reported by the Ministry of Economy, Trade and Industry and Tokyo Shoko Research (23): The University of Tokyo, Keio University, Kyoto University, Osaka University, University of Tsukuba, and Tohoku University. We revised the opening of Section 5 to clarify the source and rationale for selecting these six universities and to emphasize that this university-level assessment is conceptually separate from the policy-context description in Section 3.

• Line 497

This university-level assessment is intended as a supplementary application of the identified characteristics and is conceptually separate from the policy-context description in Section 3, where four major national universities were mentioned as an illustrative example of the Public-Private Innovation Program. Accordingly, the mention of those four universities does not imply that the startup-level analytical sample was restricted to any specific subset of universities.

(3) One-to-one mapping between startups and researchers; handling multiple researchers per startup.

Our dataset includes startups that can be associated with multiple researchers (194 startups and 252 involved researchers). For model training and performance evaluation, we did not aggregate researcher indicators within a startup (e.g., by simple or weighted averages). Instead, we constructed one-to-one startup

---

## [Decision Letter · Decision Letter 2]

16 Mar 2026

Identifying researcher characteristics driving growth in Japanese university-originated deep-tech startups: A machine learning approach

PONE-D-25-39531R2

Dear Dr. Mizuhara,

We’re pleased to inform you that your manuscript has been judged scientifically suitable for publication and will be formally accepted for publication once it meets all outstanding technical requirements.

Kind regards,

Lutz Bornmann

Academic Editor

PLOS One

Additional Editor Comments (optional):

Reviewers' comments:

Reviewer's Responses to Questions

**Comments to the Author**

Reviewer #2: All comments have been addressed

2. Is the manuscript technically sound, and do the data support the conclusions?

Reviewer #2: Yes

3. Has the statistical analysis been performed appropriately and rigorously?

Reviewer #2: Yes

4. Have the authors made all data underlying the findings in their manuscript fully available?

Reviewer #2: Yes

5. Is the manuscript presented in an intelligible fashion and written in standard English?

Reviewer #2: Yes

Reviewer #2: I want to thank the authors for their conscientious consideration of the comments I made to the preveous version of the paper. They have addressed all my issues and concerns to my satisfaction and I suggest the paper to be accepted for publication PLOS One. I very much appreciate the efforts made by the authors and I hope they feel positive about the comments made and the issues raised. Frmo my point of view the paper gained in clarity and substance.

.

Reviewer #2: No

---

## [Editor Report · Acceptance letter]

PONE-D-25-39531R2

PLOS One

Dear Dr. Mizuhara,

I'm pleased to inform you that your manuscript has been deemed suitable for publication in PLOS One. Congratulations! Your manuscript is now being handed over to our production team.

Kind regards,

on behalf of

Dr. Lutz Bornmann

Academic Editor

PLOS One